# Dimeric transport mechanism of human vitamin C transporter SVCT1

Takaaki A. Kobayashi [1], Hiroto Shimada[1,6], Fumiya K. Sano [1], Yuzuru Itoh [1], Sawako Enoki[2], Yasushi Okada [2,3,4,5], Tsukasa Kusakizako [1] ✉ & Osamu Nureki [1] ✉

Vitamin C plays important roles as a cofactor in many enzymatic reactions and as an antioxidant against oxidative stress. As some mammals including humans cannot synthesize vitamin C de novo from glucose, its uptake from dietary sources is essential, and is mediated by the sodium-dependent vitamin C transporter 1 (SVCT1). Despite its physiological significance in maintaining vitamin C homeostasis, the structural basis of the substrate transport mechanism remained unclear. Here, we report the cryo-EM structures of human SVCT1 in different states at 2.5–3.5 Å resolutions. The binding manner of vitamin C together with two sodium ions reveals the counter ion-dependent substrate recognition mechanism. Furthermore, comparisons of the inward-open and occluded structures support a transport mechanism combining elevator and distinct rotational motions. Our results demonstrate the molecular mechanism of vitamin C transport with its underlying conformational cycle, potentially leading to future industrial and medical applications.

Vitamin C (ʟ-ascorbic acid) is an essential nutrient for humans. It is required in numerous biological processes, including collagen biosynthesis and histone demethylation as a cofactor for α-ketoglutarate-dependent dioxygenases, and also acts as an antioxidant against oxidative stress[1–4]. Severe vitamin C deficiency leads to the development of scurvy, a major cause of death among sailors centuries ago[3]. Despite its biological significance, some mammals including humans are not able to biosynthesize vitamin C because of the inactivation of ʟ-gulonolactone oxidase, which catalyzes the last step of vitamin C biosynthesis[5], making the dietary intake of this nutrient indispensable for these animals.

Two families of membrane transporters are involved in vitamin C uptake and distribution: glucose transporters (GLUTs) and sodium-dependent vitamin C transporters (SVCTs). GLUTs mediate the absorption of dehydroascorbic acid (DHA), the oxidized form of vitamin C, through facilitated diffusion in competition with glucose[6,7]. GLUTs-mediated DHA uptake is important in SVCT-lacking cell types such as erythrocytes and neutrophils, where DHA is immediately reduced to ascorbic acid to serve as an antioxidant[8,9]. By contrast, SVCTs are secondary active transporters that mediate the uptake of the reduced form of vitamin C, driven by the Na⁺ electrochemical gradient[10]. Since their isolation in 1999, extensive studies have focused on the functional aspects of SVCTs. Two SVCTs are known to co-transport Na⁺ and vitamin C in a 2:1 stoichiometry[11,12], with different functional properties to cooperatively maintain vitamin C homeostasis. SVCT1 (encoded by the SLC23A1 gene) is a high-capacity transporter mainly expressed in the epithelial cells of organs such as intestine and kidney, and is responsible for maintaining the whole-body vitamin C level[13,14]. The other transporter, SVCT2 (encoded by the SLC23A2 gene), is a widely expressed high-affinity transporter that plays a role in distributing vitamin C to cells in response to oxidative stress[13,14]. Although therapeutic uses of vitamin C still remain controversial[15,16], controlling the plasma level of vitamin C is potentially effective for developing new cancer treatments[4,17], in which the functional modulation of SVCTs could play key roles.

[1]Department of Biological Sciences, Graduate School of Science, The University of Tokyo, Tokyo, Japan. [2]Department of Physics, and Universal Biology Institute (UBI), Graduate School of Science, The University of Tokyo, Tokyo, Japan. [3]Department of Cell Biology, Graduate School of Medicine, The University of Tokyo, Tokyo, Japan. [4]Laboratory for Cell Polarity Regulation, RIKEN Center for Biosystems Dynamics Research (BDR), Osaka, Japan. [5]International Research Center for Neurointelligence (WPI-IRCN), The University of Tokyo, Tokyo, Japan. [6]Present address: Research Division, Chugai Pharmaceutical Co., Ltd., Kanagawa, Japan. ✉e-mail: kusakizako@bs.s.u-tokyo.ac.jp; nureki@bs.s.u-tokyo.ac.jp

SVCTs belong to the nucleobase-ascorbate transporter (NAT) family[18], with architectures elucidated by the crystal structures of the bacterial uracil transporter UraA[19,20] and the fungal purine transporter UapA[21]. These structures revealed the fold of 14 transmembrane helices (TMs) constituting the core and gate domains, where relative domain movements play an important role in substrate transport. While various possible transport mechanisms[22], such as rocking bundle[19], elevator[21], or their combination[20], were suggested, the precise mechanism has yet to be established. Although the relationships between dimerization and transport[20,21], and lipid binding and dimerization[23] have also been suggested, the molecular mechanisms have remained enigmatic due to the limited structural information. Recently, cryo-EM structures of mouse SVCT1 were reported and a sodium-dependent substrate recognition mechanism was suggested[24]. However, these structures only showed the inward-open state, which has already been reported for homologs[19,21].

Here, we present the structures of human SVCT1 in multiple states, including the substrate-bound and substrate-free inward-open states, and the substrate-free occluded state, at 2.5–3.5 Å resolutions. Our structures clearly illuminate the vitamin C recognition mechanism of SVCT1, with updates from the recent report[24]. Comparisons of our structures in distinct states, along with biochemical analyses, provide significant insights into the dimer-based conformational changes and transport mechanism.

## Results

### Structure determination

To obtain a stable and homogenous human SVCT1 (hSVCT1) sample suitable for cryo-EM analysis, we first screened solubilization conditions for hSVCT1. The C-terminally GFP-fused hSVCT1, with functional properties equivalent to those reported previously[13] (Fig. 1a, b), was transiently expressed in HEK293T adherent cells and solubilized with various detergents, and then the degrees of stability and monodispersity were examined by fluorescent-detection size-exclusion chromatography (FSEC)[25] based on the GFP fluorescence. hSVCT1 exhibited two main elution peaks (Supplementary Fig. 1a), likely corresponding to different oligomeric forms, especially monomers and dimers, as expected from previous research[19,20,26]. Since the importance of dimer formation for the transport activity has been reported for the bacterial homolog UraA[20] and fungal homolog UapA[21], and larger molecular weights and more symmetrical properties of the target are generally advantageous in cryo-EM single particle analysis, we focused on the first peak possibly corresponding to the dimeric form, which predominantly appeared in the lauryl maltose neopentyl glycol supplemented with cholesteryl hemisuccinate (LMNG-CHS)-solubilized conditions. We expressed hSVCT1-GFP in HEK293S GnTI⁻ cells and purified it with GFP nanobody-immobilized resins, followed by size-exclusion chromatography in the presence of Na⁺ (Supplementary Fig. 1b, c). Purified hSVCT1 preparations with or without vitamin C (sodium ascorbate) were subjected to cryo-EM single particle analysis, and we determined the inward-open structures of hSVCT1 in the substrate-bound and substrate-free states at overall resolutions of 2.5 and 2.6 Å, respectively (Fig. 1c and Supplementary Figs. 2 and 3). The two structures have no significant differences except for the substrate binding, and are aligned with a root mean square deviation (RMSD) of 0.28 Å for 1,024 Cα atoms. Therefore, with regard to the inward-open conformation, we describe the structural features of the well-resolved substrate-bound state for simplicity, unless otherwise specified. We also prepared the hSVCT1 sample purified in the absence of Na⁺ (Supplementary Fig. 1d–f), and obtained its 2.8 Å resolution occluded structure by cryo-EM analysis (Fig. 1d and Supplementary Fig. 4). The details and comparisons of this novel structure will be discussed in a later section.

### Overall structure of hSVCT1 in the inward-open state

hSVCT1 forms a homodimer, with each protomer consisting of 14 transmembrane helices (TMs) organized into a "core domain" (TMs

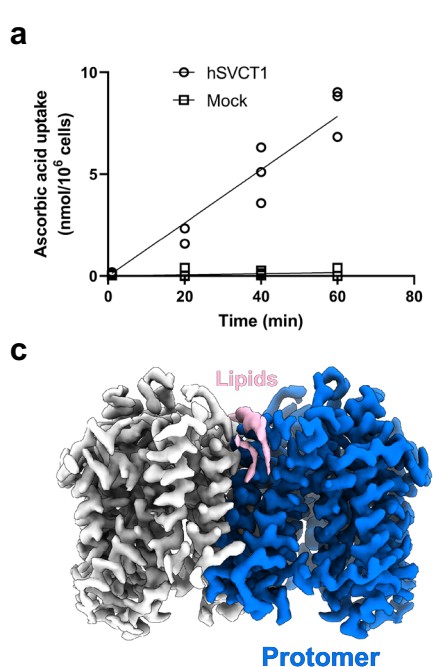

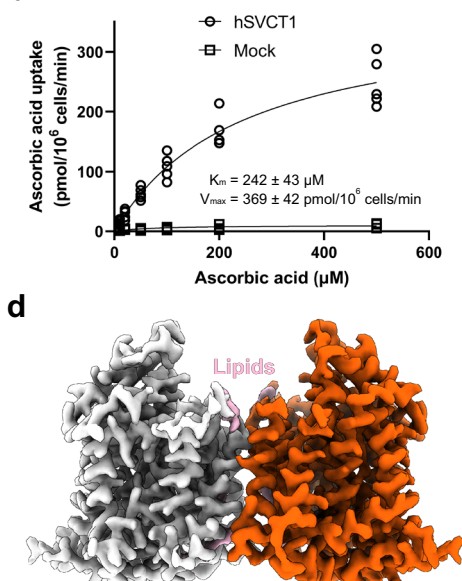

**Fig. 1 | Functional characterization and structure determination of hSVCT1.**
**a** Time-dependent ascorbic acid transport into HEK293T cells by hSVCT1 ($n = 3$ biological replicates). **b** $K_m$ curve for hSVCT1-derived ascorbic acid uptake into HEK293T cells ($n = 5$ for hSVCT1, $n = 3$ for mock). Source data are provided as a Source Data file. **c** Cryo-EM map of the hSVCT1 homodimer in the inward-open state, with densities corresponding to each protomer and lipids colored blue, gray, and pink, respectively. **d** Cryo-EM map of the hSVCT1 homodimer in the substrate-free occluded state, with densities corresponding to each protomer and lipids colored orange, gray, and pink, respectively.

In panel b: $K_m = 242 \pm 43\ \mu M$, $V_{max} = 369 \pm 42\ pmol/10^6\ cells/min$

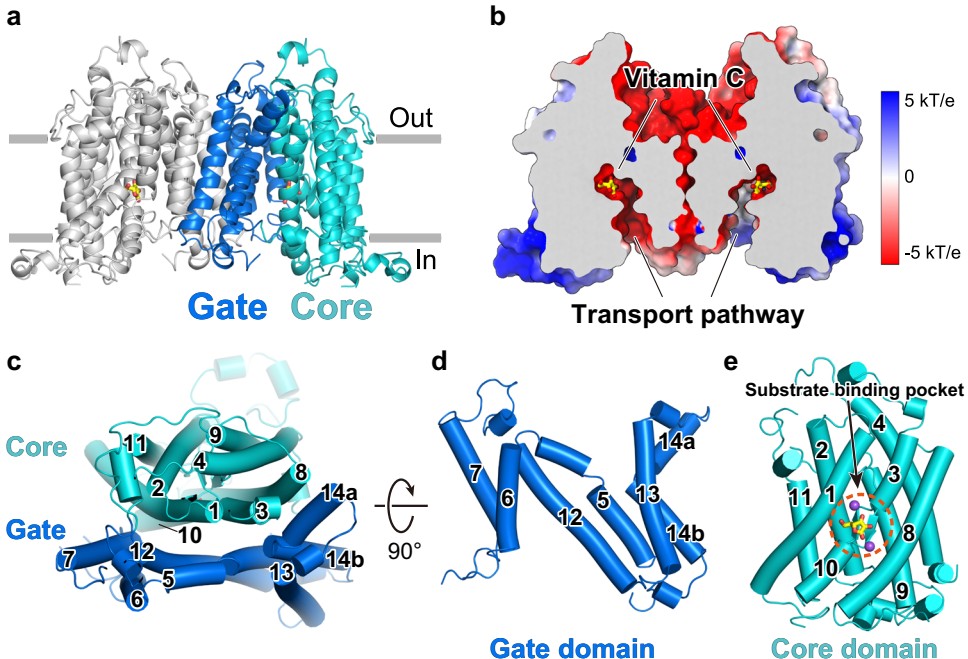

**Fig. 2 | Structure of hSVCT1 in the inward-open state. a** Overall structure of inward-open hSVCT1. The core and gate domains of a protomer are colored cyan and blue, respectively. **b** Electrostatic surface representation of inward-open hSVCT1 sliced at the transport pathway. Vitamin C bound within the substrate binding pocket is shown as yellow sticks. **c** Protomer viewed from the extracellular side, with each TM numbered. **d** Gate domain and **e** core domain viewed from the other protomer.

1–4 and 8–11) and a "gate domain" (TMs 5–7 and 12–14) (Fig. 2a, c–e and Supplementary Fig. 5). This protein fold of the protomer is referred to as the "UraA fold", and is conserved among the SLC4, 23 and 26 transporter families[19–21,27–30]. The surface model showed that it adopts an inward-open state with a conserved substrate binding pocket accessible from the cytoplasmic side (Fig. 2b). A cleft is formed at the interface of the core and gate domains, and the substrate binding pocket is located at the end of the cleft, backed by a pair of short antiparallel β-strands between two half-helices of TMs 3 and 10 (Fig. 2e). Considering the similarity of the two obtained inward-open structures, we assumed that the substrate-bound and substrate-free structures correspond to the states before and after substrate release, respectively.

The formation of the hSVCT1 dimer is predominantly achieved by interactions between the TM regions in a similar manner to its homologs[20,21], where the gate domains face each other with an extensive interface of ~2,000 Å², partly mediated by lipids (Fig. 1c and Supplementary Fig. 6). This is in stark contrast to the SLC4 and 26 transporters, where the gate domains form either a smaller or no interface[27,29]. The SLC4 and 26 transporters have a large cytosolic domain that connects two protomers[29,31,32], while hSVCT1 has a small structural component outside the plasma membrane, suggesting that the absence of a cytosolic domain in the SLC23 family, including SVCT1, facilitates the formation of a predominant dimeric interface between the TM domains. An N-linked glycan is present at Asn144 on the extracellular loop connecting TMs 3 and 4 (Supplementary Fig. 6g), where *N*-glycosylation is reportedly important for membrane targeting and substrate transport[33]. In addition, *S*-acylation is observed at Cys565, located in the protruding loop region at the border between the plasma membrane and cytoplasm (Supplementary Fig. 6h). Although its function has not been reported so far, the *S*-acylation of this residue, with its attached fatty acid chain inserted within the membrane, may contribute to protein stability or membrane trafficking[34].

## Vitamin C recognition mechanism

We identified a vitamin C molecule bound at the conserved substrate binding site inside the protomer. Our density map is clear enough to reliably determine its orientation (Fig. 3a and Supplementary Fig. 7a). Several non-protein densities are observed at the pocket in addition to vitamin C, and we modeled two sodium ions and three water molecules according to the coordination of the surrounding residues (Fig. 3b–d). The presence of two sodium ions together with one vitamin C molecule is consistent with previous reports that SVCTs transport sodium ions and vitamin C in a 2:1 stoichiometry[11,12,35]. As the substrate binding pocket is potentially negatively charged (Fig. 2b), sodium ions are required to neutralize the negative charge to facilitate vitamin C binding, confirming the proposed mechanism reported for the mouse SVCT1 structure[24]. Additionally, TMs 3 and 10 sandwich vitamin C, and their dipole moments electrostatically stabilize the bound substrate, as observed in the Cl⁻ binding of Prestin[30]. The first sodium ion, Na1, is surrounded by TMs 8 and 10, as well as vitamin C. Na1 interacts with the sidechains of Glu334, Asp338, and Ser383, the backbone carbonyl of Ser381, and an oxygen atom of vitamin C (Fig. 3b). The location and coordination manner of Na1 are generally consistent with those in mouse SVCT1[24] and similar to the sodium ion binding mode of the SLC4 transporter NDCBE[36]. A notable feature in our structure is the direct interaction between Glu334 and vitamin C, which emphasizes the importance of the conserved glutamate residue in substrate recognition. A comparison between the substrate-bound and substrate-free structures revealed that the sidechain of Ser383 is flipped after releasing the substrate and directly interacts with the sidechain of Asp338, without mediation by a sodium ion, suggesting that substrate release triggers the reorganization of the substrate binding pocket, and possibly induces the subsequent conformational change (Supplementary Fig. 7b). The second sodium ion, Na2, is located opposite Na1 across the vitamin C, where strong density extending from the substrate is observed (Fig. 3a, c and Supplementary Fig. 7a). Na2 is stabilized by a cation-π interaction with the aromatic sidechain

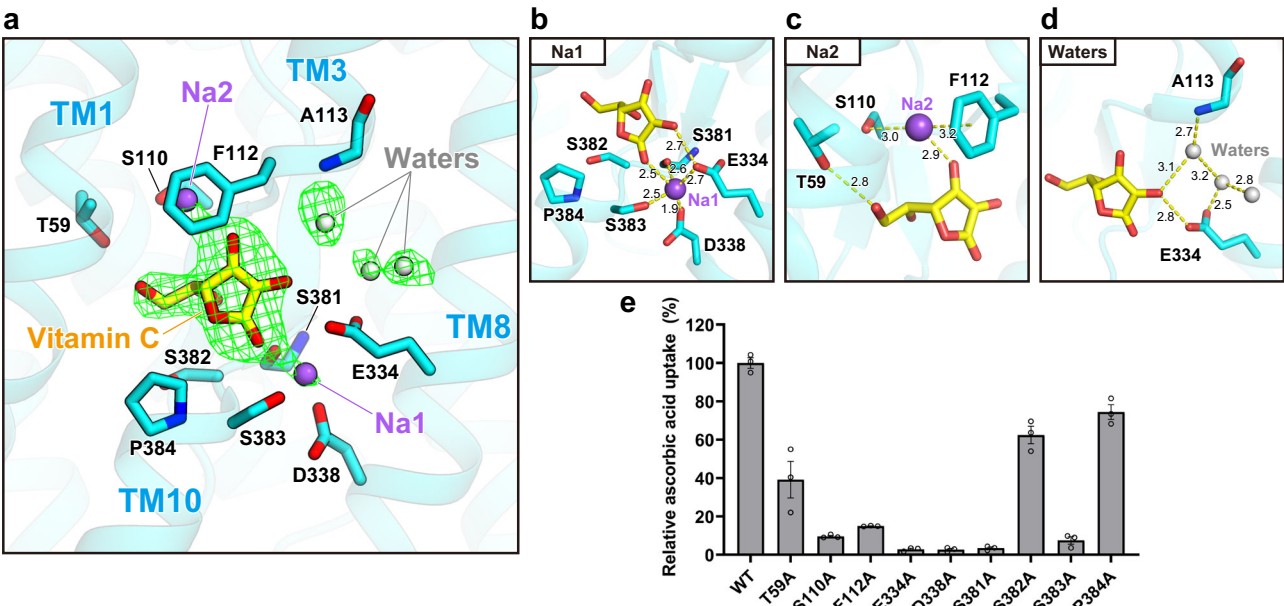

**Fig. 3 | Substrate coordination manner. a** Overall view of the substrate binding pocket. The green mesh represents the $F_o$-$F_c$ omit map of vitamin C, two sodium ions, and three water molecules contoured at 3.0σ (where σ is the standard deviation within the mask). **b**–**d** Close-up views and interaction manners of each ligand. **e** Substrate uptake assays for point mutants. Values are mean of $n = 3$ biological replicates ± s.e.m. and compared relative to the wild-type uptake rate. Source data are provided as a Source Data file.

of Phe112, as well as by polar interactions with Ser110 and vitamin C. Notably, the position of Na2 in our structure is different from that in the mouse SVCT1 structure, in which Na2 is surrounded by TMs 3 and 8 on the same side as Na1 (ref. 24; Supplementary Fig. 8). Although non-protein density is also observed at that position in our density map, we modeled a water molecule based on the hydrogen bonds from vitamin C and the backbone amide of Ala113 (Fig. 3d). Under neutral pH conditions, vitamin C exists as a monovalent anion in resonance between two forms, with a negative charge on the oxygen atom covalently bonded to either the C1 or C3 carbon[2,37] (Supplementary Fig. 8). In our structure, two sodium ions are located beside these potentially negatively charged oxygen atoms, thus confirming the stable coordination manner of the positively charged ions. Together with the two sodium ions and three water molecules, the formation of a broad hydrogen bond network around the vitamin C is likely to play a role in stabilizing the substrate binding. Another interaction is the polar interaction between Thr59 and the oxygen atom of the vitamin C's 1,2-dihydroxyethyl group (referred to as the "tail" for simplicity; Fig. 3c and Supplementary Fig. 9). While the lactone ring of vitamin C is strongly stabilized by interactions with the core domain through two sodium ions and water molecules, the tail forms relatively few interactions with the protein chain and neighboring small molecules. This biased distribution of interactions with the substrate possibly contributes to the incomplete selectivity for vitamin C and its stereoisomer, where L-ascorbic acid is favorably transported and D-isoascorbic acid, which only differs in the orientation of the tail, is transported less efficiently but in detectable amounts[10,38].

To examine the functional importance of the residues comprising the substrate binding pocket, we measured the transport activities of hSVCT1 mutants by a cellular ascorbic acid uptake assay, based on its reducing ability (Fig. 3e). The results demonstrated that the alanine mutations of the residues involved in the coordination of two sodium ions significantly decreased the transport activity, confirming the sodium-dependent binding manner of the substrate suggested from the structure. The mutation of Thr59, which interacts with the tail of the vitamin C molecule, caused a partial impairment of substrate uptake, consistent with the proposed weak recognition of this region.

Although we did not detect any direct interactions between the substrate and residues such as Ser382 and Pro384, mutants of these residues exhibited rather decreased transport activities. These residues may have a role in substrate affinity by forming and stabilizing the hydrophilic pocket.

## hSVCT1 structure in the substrate-free occluded state

To elucidate the molecular mechanism of the overall transport cycle, other conformations of the hSVCT1 structure must be solved. Focusing on the sodium-dependent transport by SVCTs, we assumed that hSVCT1 would adopt different transporting conformations under sodium-free conditions. After confirming that hSVCT1 exhibits a similar and stable oligomerization profile with the substitution of sodium with potassium in all buffers, we purified hSVCT1 dimers using the same procedure as for the sodium-containing conditions, except for the switching of cations (Supplementary Fig. 1d–f). The cryo-EM analysis, performed using the same procedure as for the sodium-containing conditions, resulted in two major 3D reconstruction classes (classes 1 and 2; Supplementary Fig. 4). Whereas class 1 showed a similar conformation to that of the inward-open states under sodium-containing conditions, class 2 showed a different conformation, whose particles resulted in a final refinement at 2.8 Å resolution (Figs. 1d and 4a and Supplementary Fig. 4). The substrate binding pocket in this map is vacant, and sequestered from the solvent mainly by the side-chains of Phe112, Glu334, and Ser383, suggesting that this structure represents a substrate-free occluded state (Fig. 4b, c). This substrate-free occluded structure reveals an unprecedented conformational state for the NAT family, representing a structure between the substrate-free inward-open and outward-open states. The occluded substrate binding pocket may be too small to accommodate a vitamin C molecule, indicating its different structural properties from the substrate-bound occluded structure previously reported for UraA[20] (Supplementary Fig. 10). For the inward open-like class under the potassium conditions (class 1), the final reconstruction resulted in a density map at a resolution of 3.5 Å (Supplementary Figs. 4 and 11). The intracellular region of the core domain, especially TM10, is disordered in this structure, suggesting the flexibility of this region. This class

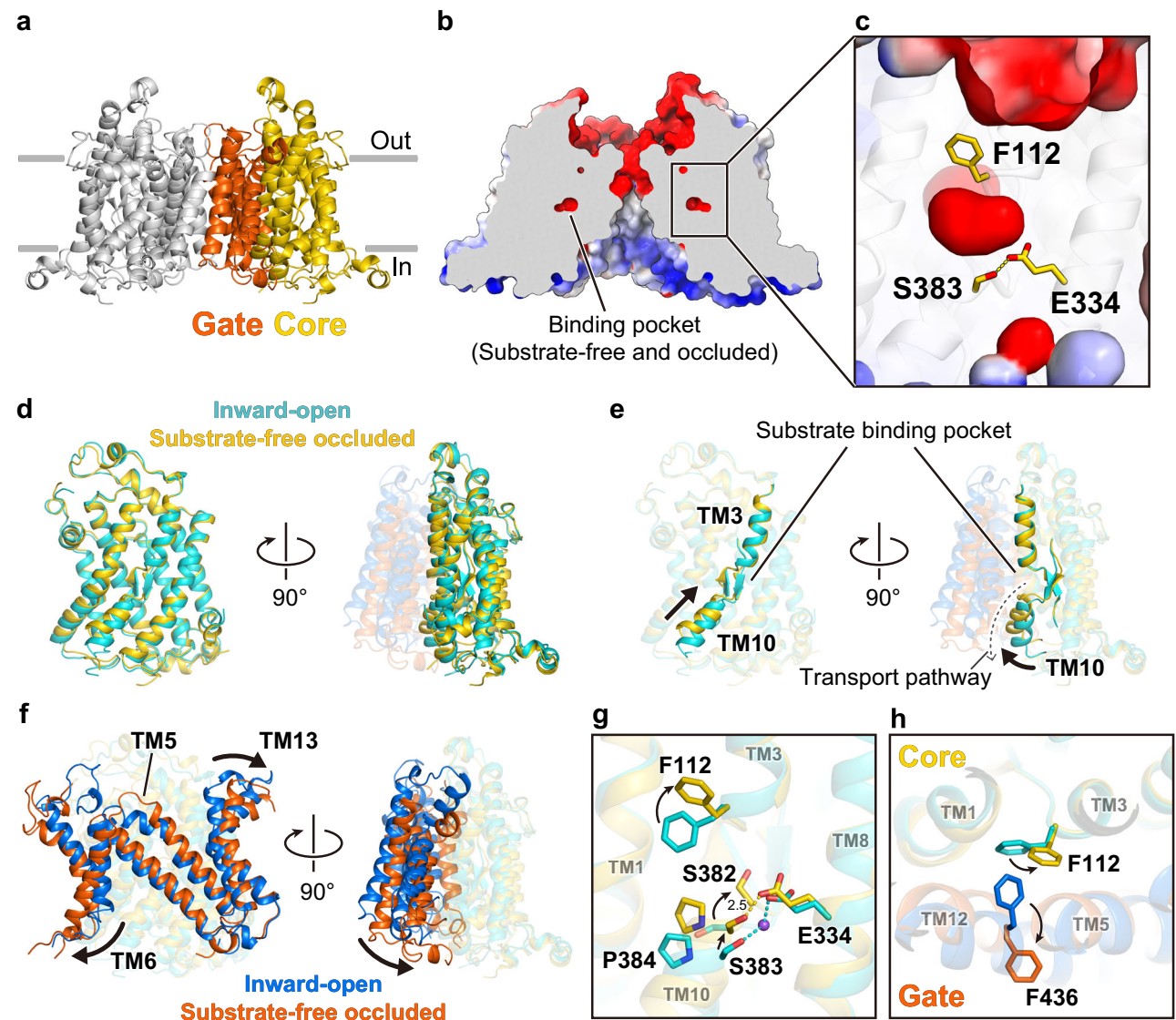

**Fig. 4 | hSVCT1 in the substrate-free occluded state. a** Overall structure of substrate-free occluded hSVCT1. The core and gate domains of a protomer are colored yellow and orange, respectively. **b** Electrostatic surface representation of substrate-free occluded hSVCT1 sliced at the substrate binding pocket. **c** Close-up view of the occluded pocket. **d**, **e** Comparison of the core domains between inward-open and substrate-free occluded hSVCT1, aligned by the protomeric core domain. **(d)** Overall alignment of the core domain and **(e)** highlights of TMs 3 and 10. **f** Comparison of the gate domain aligned by the core domain of the same protomer. **g**–**h** Comparison of the residues comprising **(g)** the substrate binding pocket and **(h)** the extracellular gate.

possibly corresponds to an intermediate state during the transition from the inward-open state to the substrate-free occluded state, indicating the conformational equilibrium between two distinct states.

A superimposition of the core domains of the inward-open and substrate-free occluded structures revealed a good alignment with an RMSD of 1.32 Å for 308 Cα atoms with no significant differences, except for the notable upward shift of TM10 towards the extracellular side from the inward-open to substrate-free occluded state (Fig. 4d, e). The up-and-down movements of the core domain, especially for TMs 3 and 10, were reported in previous studies of SLC proteins structurally related to NAT transporters[29,30,32,39,40], and are referred to as elevator movements. UapA[21] and mouse SVCT1[24] also adopt this elevator mechanism through the transition from the substrate-bound occluded to inward-open states, with TMs 3 and 10 shifting downward to the intracellular side. The movement captured here can also be regarded as an example of elevator-like movement, but a notable difference from the previous reports is the additional horizontal shift of TM10 toward the gate domain, which occludes the transport pathway and

prevents access to the interior of the protein (Fig. 4e). Flexibility of TM10 observed in the intermediate structure also supports the large movement of this region (Supplementary Figs. 5c and 11). The residues comprising the substrate binding pocket, such as Ser382, Ser383, and Pro384, are shifted to the extracellular side along with the movement (Fig. 4g). The sidechains of Glu334 and Ser383, which are responsible for the coordination of Na1 in the substrate-bound inward-open state, are in close contact and directly interact without a sodium ion in this occluded state, as Asp338 and Ser383 in the substrate-free inward-open state, contributing to the formation of the intracellular gate (Fig. 4c, g and Supplementary Fig. 7b). Ser382, whose sidechain is not involved in the substrate recognition, is flipped and oriented towards the substrate binding pocket after the conformational change, suggesting the importance of this residue for the occlusion of the pocket after substrate release (Fig. 4g).

Superimposition of the protomers of the inward-open and substrate-free occluded structures relative to the core domain revealed that the entire gate domain tilts toward the cytoplasmic side

of the core domain in the same protomer, with TMs 6 and 13 outwardly rotated in the transition from the inward-open state to the substrate-free occluded state (Fig. 4f). This tilting motion of the gate domain is likely related to the rocking bundle-like mechanism proposed in previous studies of UraA monomer and dimer structures[19,20]. We observed the flipping of the aromatic sidechains of Phe112 in the core domain and Phe436 in the gate domain through the conformational change (Fig. 4g, h). In the inward-open state, Phe112 forms a π-π stacking interaction to constitute the extracellular gate, while Phe112 also coordinates Na2 by a cation-π interaction on the opposite side of the phenyl group (Fig. 3c). In the substrate-free occluded state without a sodium ion, however, the stacking is disrupted due to the flipping of both phenylalanine residues and the extracellular gate is formed only by Phe112, indicating the fragility of the gate in this occluded state

(Fig. 4c, h). Thus, this state possibly represents a state just prior to switching to the outward-open state and importing the next substrate.

## Dimer formation is requisite for transport function

A comparison of the dimer interfaces between the inward-open and substrate-free occluded structures revealed an extensive conformational change (Fig. 5a, b and Supplementary Movie 1). In the inward-open state, a stable dimer interface is formed mainly by TMs 6, 12, and 13 of the gate domains, with TM13 inserted into a cleft on the intracellular side formed by TMs 6′ and 12′ of the other protomer (apostrophe indicates the adjacent protomer; Fig. 5a). However, in the substrate-free occluded state, the gate domains are reorganized to form a different dimer interface, in which TM13 has left the cleft and become exposed to the plasma membrane (Fig. 5b). Detailed obser-

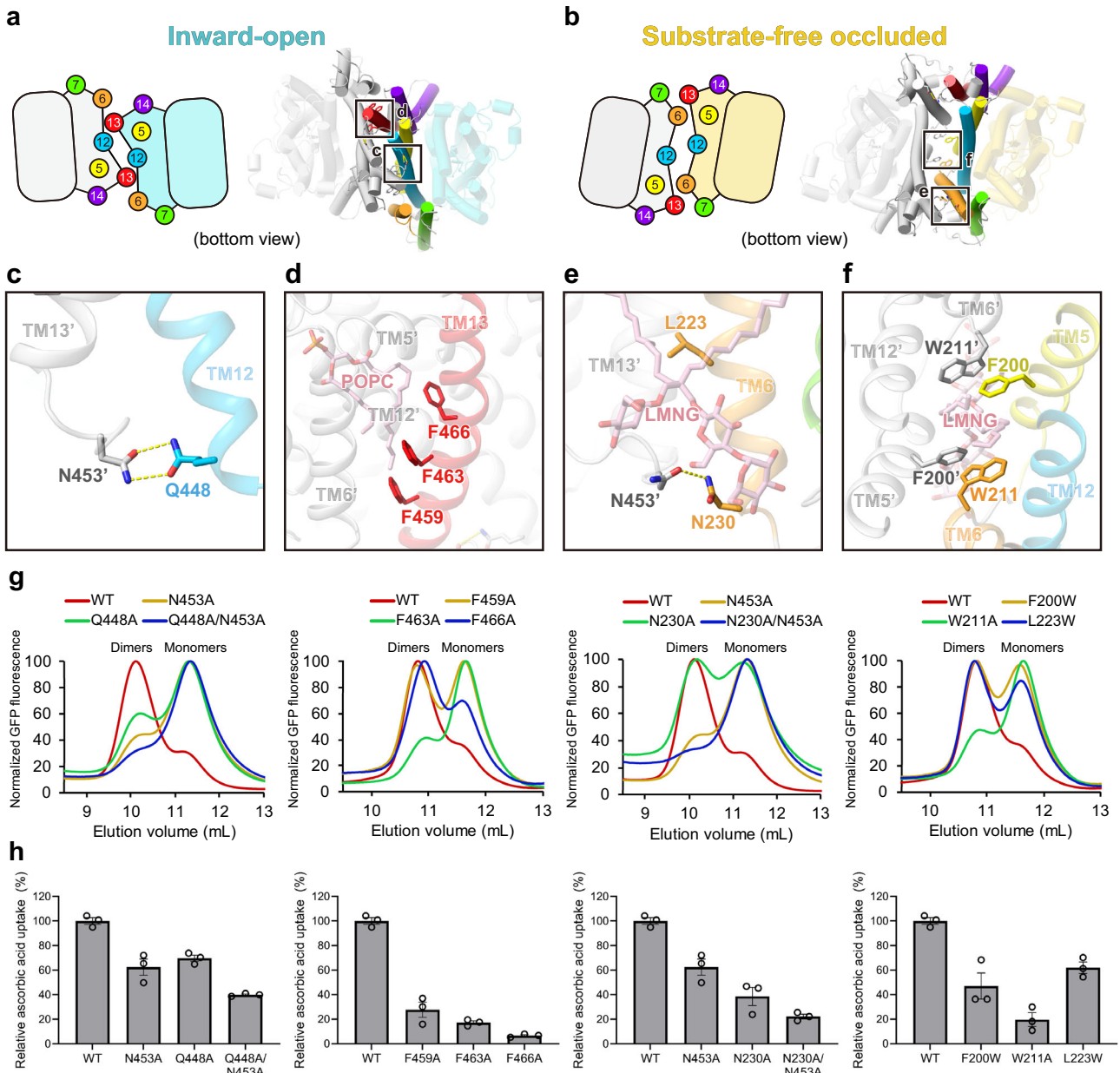

**Fig. 5 | Dimer formation and transport activity of hSVCT1. a–b** Representations of dimer interfaces in (**a**) inward-open state and (**b**) substrate-free occluded state, viewed from the intracellular side. **c–f** Close-up views of the residues responsible for dimer formation in (**c–d**) inward-open state and (**e–f**) substrate-free occluded state. **g** Evaluation of oligomeric population of dimer interface mutants by FSEC, detected by GFP fluorescence. **h** Substrate uptake assay for dimer interface mutants. Values are mean of $n = 3$ biological replicates ± s.e.m. compared relative to the wild-type uptake rate. Source data are provided as a Source Data file.

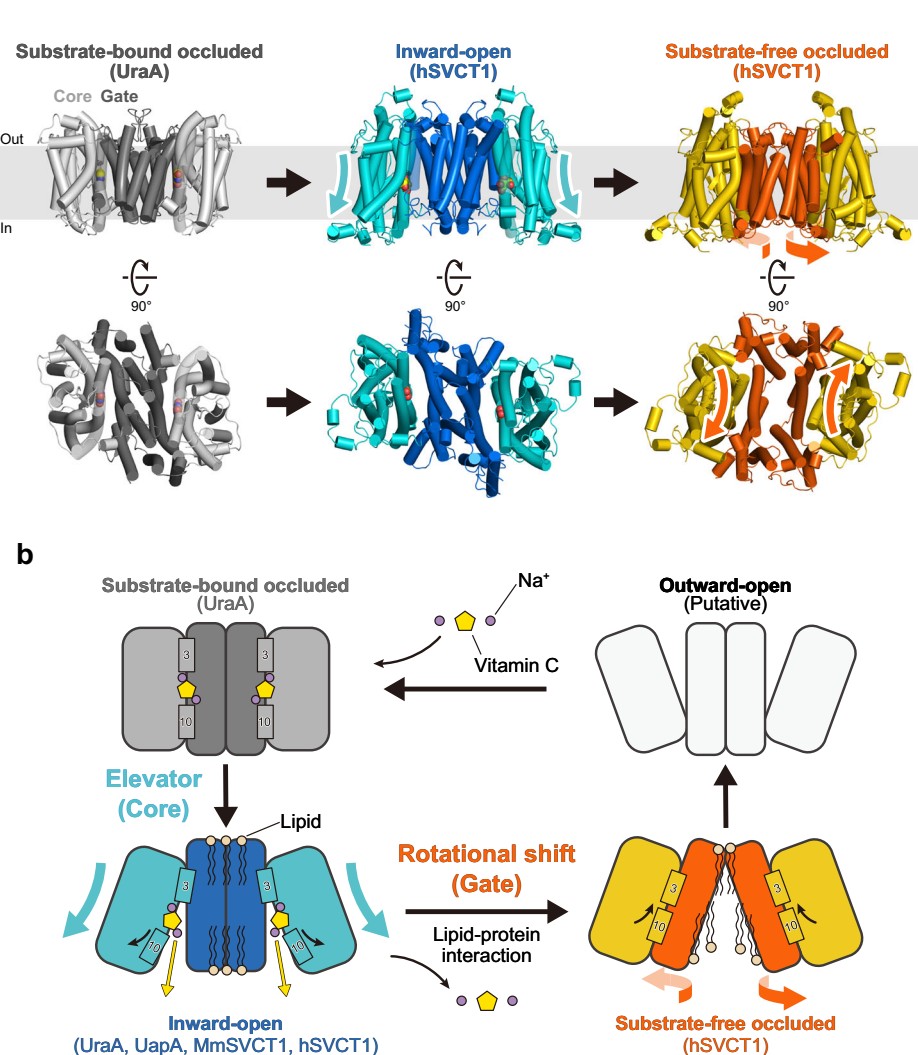

**Fig. 6 | Dimeric structural changes underlying the transport cycle.**
**a** Comparison of dimer structures of hSVCT1 and UraA between three states, in the order of putative substrate transport. Colored arrows indicate the relative movements of domains. **b** Schematic model for the predicted transport cycle of NAT/ SLC23 members. Experimental structures are determined for the three colored states, and no structural information is currently available for the uncolored outward-open state.

vations of the dimer interfaces of the two states revealed the rearrangements of hydrogen bonds and the state-specific hydrophobic interactions between protomers. As for hydrogen bonds, Asn453' on TM13' forms a bipartite hydrogen-bond with Gln448 on TM12 on the intracellular side in the inward-open structure (Fig. 5c), while Asn453' on TM13' hydrogen-bonds with Asn230 on TM6 in the occluded structure, to stabilize the dimer formation in each state (Fig. 5e). As for hydrophobic interactions, multiple phenylalanine sidechains on TM13 are inserted into the cleft between TM6' and TM12' to form the unique dimer interface in the inward-open state, accompanied by a lipid molecule (Fig. 5d). In the occluded state, Trp211 on TM6, which faced outwardly from the protein in the inward-open state, has turned inside and interacts with Phe200' on TM5' to stabilize the formation of a differently organized dimer interface together with an LMNG molecule (Fig. 5f). In addition, Leu223 on TM6 is located adjacent to another LMNG molecule at the dimer interface, coordinating the close packing of the interface in the occluded structure (Fig. 5e).

To investigate the importance of these residues in the distinct dimer interactions and the transport activity of hSVCT1, we introduced mutations of these residues to diminish the interactions between protomers or overlap with the lipid molecule. We examined the ratio of monomeric or dimeric hSVCT1 population by FSEC, and found that the

FSEC peak partially shifted from the dimeric to monomeric position for all mutants (Fig. 5g). Notably, mutants of hydrogen-bonded residue pairs exhibited a gradual decrease in dimer formation as more mutations were introduced for both the Gln448/Asn453 and Asn230/ Asn453 pairs, supporting the importance of these residues for stabilizing the dimerization and the existence of two states under physiological conditions. We tested the transport activities of these mutants, and found that vitamin C transport decreased for all mutants in accordance with the ratio of dimeric populations, except for F466A, suggesting the relationship between the substrate transport activity and the formation of two distinct dimeric states (Fig. 5h). The F466A mutation did not inhibit dimer formation as much as other mutations, but its transport activity was quite low (Fig. 5g–h), indicating a crucial role of Phe466 in substrate transport. This mutation may affect the dimer interface rearrangement, rather than the dimer formation. The proper membrane expression of these hSVCT1 mutants was confirmed by observing fused GFP fluorescence using confocal microscopy (Supplementary Fig. 12), suggesting that destabilization of dimer formation do not affect trafficking of hSVCT1 to the plasma membrane, similarly to the fungal homolog UapA[41]. As a number of studies have provided evidence for the relationship between dimer formation and protein function of NAT family members[20,21,23,42], these results further

suggest the necessity of two distinct functional dimers to rearrange the interface into differently organized conformations through the transport cycle.

## Structural comparison and conformational changes of dimers

To gain insights into the conformational changes of the dimer during the transport cycle, we compared our structures of hSVCT1 in two states with the previously reported dimeric UraA structure[20], which is regarded as a representative of the substrate-bound occluded state of NAT/SLC23 family transporters, in accordance with the order of the predicted substrate transport cycle (Fig. 6a, Supplementary Fig. 13). When comparing the UraA structure in the substrate-bound occluded state with the hSVCT1 structure in the inward-open state, superimposed relative to the dimeric gate domains, the gate domains showed no significant movement between the two states, and thus probably maintain a stable dimer interface in this transition (Supplementary Fig. 13a). In contrast, the core domains exhibited a significant translocation toward the cytosolic side against the gate domains, with opening the substrate transport pathway between the core and gate domains inside each protomer (Supplementary Fig. 13b and Supplementary Movie 1). The bound substrate also shifts towards the cytosolic side through this transition (Supplementary Fig. 13a). This movement of the core domains is consistent with the typical elevator mechanism, as suggested in related structural reports[21,24].

In stark contrast, a comparison of our two hSVCT1 structures in the inward-open and substrate-free occluded states, superimposed relative to the core domains, revealed the distinct movement of the gate domain. In addition to the tilting movement observed in the protomeric comparison (Fig. 4f), the gate domain exhibited a horizontal shift relative to the core domain of the same protomer, possibly due to the structural rearrangement of the dimer interface discussed in the previous section (Supplementary Fig. 13c). As a result of this complex movement, the gate domain undergoes a rotational shift around the core domain of the same protomer and consequently approaches the core domain to occlude the transport pathway, while widely opening the cytosolic side of the interface between the dimers (Fig. 6a and Supplementary Movie 1). Simultaneously, the core domain slightly moves upward while maintaining a relatively rigid-body structure, in addition to the large shift of TM10 described in the protomeric comparison (Supplementary Fig. 13d). This is likely an extension of the elevator movement, which is augmented by the extensive movement of the adjacent gate domain and probably contributes to the transition to the next conformational state.

Interestingly, we found some lipid-like densities in both maps of the inward-open and occluded structures. We modeled phospholipid and cholesterol molecules, as well as the LMNG and CHS of detergent molecules, based on the shapes and sizes of the densities (Supplementary Figs. 6 and 14). Lipid-protein interactions have been suggested to contribute to the functional modulation or structural stabilization of transporter oligomers[43,44]. For instance, the lipid binding by the SVCT1-related proteins, UapA and prestin, was suggested to be important for dimerization and protein function[23,29]. The lipids are mainly localized on the protein surface over the boundary of the protomers in the inward-open structure (Supplementary Fig. 6). By contrast, in the substrate-free occluded structure, the widely opened inter-protomer cavity is occupied with multiple LMNG and CHS molecules, which seem to compress the gate domains (Supplementary Fig. 14). As the existence of lipid molecules at the dimer interface is consistent with the prediction for UapA in which lipids play a key role in the formation of functional dimers[23], these LMNG and CHS molecules possibly mimic endogenous lipids and cholesterols, respectively. This notion is also supported by the fact that dimers are more stable under conditions containing LMNG than those containing DDM

(Supplementary Fig. 1a, d), which can be ascribed to the difficulty of mimicking a phospholipid with two acyl tails by DDM with its single acyl chain.

## Discussion

A major question about the transport mechanism of NAT/SLC23 family proteins is the domain motions during the transport cycle. Several hypotheses have been proposed from structural comparisons, such as the rigid-body rotation of the gate domain relative to the core domain[19], the elevator-like shift of the core domain against the dimeric gate domain[21,24], and the combination of rotations for both the core and gate domains[20], and thus there is currently no consistent view. In this study, we determined the high-resolution structures of hSVCT1 in different transporting states, which provide significant mechanistic insights into the conformational changes of NAT/SLC23 family transporters and enable more extended descriptions of the transport cycle (Fig. 6b and Supplementary Movie 1).

The transition from the substrate-bound occluded state to the inward-open state involves the elevator mechanism of the core domains, typically represented by the vertical movements of TMs 3 and 10, which enable the exposure of the substrate binding pocket to the cytosol. After substrate release, the following transition from the inward-open state to the substrate-free occluded state starts, in which the gate domains change their interactions at the dimer interface to rotationally move around the core domain, leading to the occlusion of the transport pathway to achieve the conformational change to the occluded state. We suggest that dimer interface reassembly is necessary for achieving this drastic motion of the gate domain, which probably enhances the relative movements of the domains and can explain the close relationship between dimer formation and transport activity for this family of transporters. Together with the reduced interactions at the extracellular gate, this movement also possibly leads to the following outward-facing conformation.

Lipids are observed around and inside the dimerized structures, and may play a role in the conformational transitions. It is possible that lipids on the surface of the inward-open structure slide into the dimer interface during the transition to the occluded state, driving the conformational change and stabilizing the substrate-free occluded state through lipid-protein interactions. However, as our results do not provide sufficient evidence regarding the involvement of lipids in the structural changes, further studies are still needed in this respect. We also cannot exclude the possibility that the differences between the two occluded structures are derived from the unique properties of the UraA and hSVCT1 proteins. Elucidating the substrate-bound occluded and outward-open structures of hSVCT1 itself may clarify the entire transport cycle.

Although the SLC4, 23, and 26 families share the architecture of the TM region as protomers, the dimer interface is remarkably different between the SLC4/26 and SLC23 families. While the SLC4/26 members have fewer or almost no dimer interactions between the TM regions[27,29], the SLC23 members form extensive interactions between the TM regions at the dimer interface. This characteristic formation of the SLC23 dimer interface may lead to the unique state exchange mechanism employing the distinct motions of the core and gate domains, which have not yet been observed in SLC4 or 26 family transporters.

In summary, our cryo-EM structures of hSVCT1 illuminate the precise sodium-dependent mechanism of vitamin C recognition in detail, and provide mechanistic insights into the unexpected conformational changes during the transport cycle. We believe that our findings will facilitate a deeper understanding of the molecular mechanism of substrate transport by the SLC23/NAT family, and lead to new strategies of vitamin C uptake modulation that will eventually contribute to human health.

## Methods

### Screening for solubilization conditions

The gene encoding full-length human SVCT1 (hSVCT1, SLC23A1; Uni-Prot ID Q9UHI7) was amplified from a human universal reference complementary DNA library (Zyagen) and cloned into the pEG BacMam[45] vector containing a C-terminal human rhinovirus 3C (HRV3C) protease cleavage site followed by a green fluorescent protein (GFP) and a His$_8$-tag. HEK293T cells were cultured in Dulbecco's Modified Eagle Medium (DMEM, Sigma) supplemented with 10% (v/v) fetal bovine serum (FBS, Nichirei) at 37 °C with 5% CO$_2$. Cells were seeded in 6-well culture plates (Thermo Fisher Scientific) at a density of $6 \times 10^5$ cells/mL and grown for 24 h. The cells were then transiently transfected with 1 μg of pEG BacMam vectors containing the wild-type hSVCT1 coding region and a C-terminal GFP tag, using the Lipofecta-mine 3000 transfection reagent (Invitrogen), and incubated for 48 h at 37 °C with 5% CO$_2$. Cells were harvested, washed, and resuspended in buffer containing 20 mM HEPES, pH 7.0, and 150 mM NaCl, and divided into 4 aliquots. Each aliquot was supplemented with different detergents (1% N-dodecyl-β-$_D$-maltoside (DDM, Calbiochem); 1% DDM and 0.2% (w/v) cholesterol hemisuccinate (CHS, Sigma-Aldrich); 1% lauryl maltose neopentyl glycol (LMNG, Anatrace); 1% LMNG and 0.1% CHS) and solubilized for 1 h at 4 °C. Insoluble materials in cell lysates were removed by ultracentrifugation at $108,000 \times g$ for 20 min at 4 °C, and supernatants were loaded onto a Superdex 200 Increase 10/300 GL column (GE Healthcare) attached to a fluorescence detecting monitor. The column was pre-equilibrated with buffer containing 20 mM HEPES, pH 7.0, 150 mM NaCl, and 0.03% DDM before loading the DDM- or DDM-CHS-solubilized lysate, or with buffer containing 20 mM HEPES, pH 7.0, 150 mM NaCl, and 0.01% LMNG before loading the LMNG- or LMNG-CHS-solubilized lysate. The screening under sodium-free conditions was performed using the same procedure as for the sodium-containing conditions, except for the substitution of Na$^+$ with K$^+$ in all buffers.

### Protein expression and purification

Recombinant baculoviruses of wild-type hSVCT1-GFP were generated in *Spodoptera frugiperda* Sf9 cells, using the Bac-to-Bac system (Invitrogen). HEK293S GnTI$^-$ cells were cultured in FreeStyle 293 medium (Gibco) supplemented with 2% FBS at 37 °C with 8% CO$_2$. For protein expression, P3 baculoviruses were added at a ratio of 10% (v/v) to the cell culture, at a density of approximately $3 \times 10^6$ cells/mL. To boost protein expression, sodium butyrate was added to the culture to a final concentration of 10 mM after 12 h of incubation, and the cells were further incubated for 24 h at 37 °C. Cells were collected by centrifugation at $5000 \times g$ for 12 min at 4 °C, flash-frozen in liquid nitrogen, and stored at −80 °C for later purification.

For purification, all procedures were performed at 4 °C. Cell pellets were thawed and resuspended in Buffer A (20 mM HEPES-NaOH, pH 7.0, 150 mM NaCl, 1 mM dithiothreitol (DTT), 5.2 μg/mL aprotinin, 2.0 μg/mL leupeptin, 1.4 μg/mL pepstatin and 1 mM PMSF), followed by probe sonication for 2 min. Cell debris was removed by centrifugation at $10,000 \times g$ for 10 min, and the membrane fraction was collected by ultracentrifugation at $186,000 \times g$ for 1 h. Membrane pellets were resuspended in Buffer A supplemented with 1% (w/v) LMNG and 0.1% (w/v) CHS, and homogenized with a glass Dounce homogenizer. The membrane was solubilized for 1 h, and insoluble materials were removed by ultracentrifugation at $186,000 \times g$ for 30 min. The supernatant was collected and incubated with CNBr-Activated Sepharose 4 Fast Flow beads (GE Healthcare) coupled with an anti-GFP nanobody[46] for 1.5 h. The resin was poured into an open column and washed with 10 column volumes (CV) of Buffer B (20 mM HEPES-NaOH, pH 7.0, 150 mM NaCl, 1 mM DTT, and 0.01% (w/v) LMNG). The resin was then suspended in 1 CV of Buffer B supplemented with HRV3C protease fused with a His$_{10}$-tag, and incubated overnight to cleave the GFP-His$_8$ tag. The flow-through fraction was collected and subjected to Ni-NTA affinity chromatography to remove the cleaved tag and the protease, and further purified by size-exclusion chromatography on a Superdex 200 Increase 10/300 GL column, pre-equilibrated with Buffer B. The fractions containing the hSVCT1 dimer were pooled and concentrated to 6-8 mg/mL for electron microscopy studies. To obtain substrate-bound hSVCT1, the purified protein was incubated with 1 mM sodium ascorbate for 1 h before grid preparation. The purification process under sodium-free conditions was performed using the same procedure as for the sodium-containing conditions, except for the substitution of Na$^+$ with K$^+$ in all buffers.

### Cryo-EM sample preparation and data acquisition

All cryo-EM grids were prepared by applying 3 μL of protein solution onto a freshly glow-discharged Quantifoil holey carbon grid (R1.2/1.3, Au, 300 mesh). Grids were blotted for 4 s at 4 °C under 100% humidity and plunge-frozen in liquid ethane, using a Vitrobot Mark IV (Thermo Fisher Scientific). Cryo-EM datasets were collected on a Titan Krios G4 microscope (Thermo Fisher Scientific) equipped with a K3 Summit direct electron detector (Gatan) with a BioQuantum GIF energy filter (slit width of 25 eV). Automated acquisitions were performed using the SerialEM[47] and EPU software (Thermo Fisher Scientific) for the datasets of the sodium-containing samples and the sodium-free samples, respectively. Movies were acquired at a nominal magnification of ×105,000, corresponding to a calibrated pixel size of 0.83 Å/pix with a defocus range of −0.8 to −1.6 μm, at a total dose of ~50 e$^-$/Å$^2$ over 48 frames.

### Data processing

For all datasets, image processing was performed with RELION-3.1 (ref. [48]). Beam-induced motion correction and dose weighting were performed with RELION's implementation of the MotionCor2 algorithm[49], and the contrast transfer function (CTF) parameters were estimated with CTFFIND-4.1.13 (ref. [50]).

For the substrate-bound inward-open state dataset, particles were first picked from 500 randomly selected micrographs using the Laplacian-of-Gaussian algorithm to generate 2D class average images. Using another 1,000 randomly selected micrographs, 2D reference-based auto-picking was performed and particles were subjected to several rounds of 2D and 3D classifications, resulting in the initial 3D model. For all 3,753 micrographs of the dataset, 3D reference-based auto-picking was performed, and 2,852,953 picked particles were extracted with a pixel size of 3.32 Å/pix. After several rounds of 2D and 3D classifications, 411,353 particles of the best-resolved class were re-extracted with a pixel size of 1.11 Å/pix and subjected to 3D refinement, resulting in a 3.02 Å map. Per-particle CTF refinement[51] and Bayesian polishing[52] were performed, and particles were re-extracted with a pixel size of 1.00 Å/pix. Micelles were subtracted from the particle images, and a 3D classification without alignment was performed on the subtracted particles. The selected 147,401 particles were subjected to per-particle CTF refinement and 3D refinement with C2 symmetry using the reconstruction algorithm SIDESPLITTER[53], resulting in the final map with a global resolution of 2.49 Å according to the Fourier shell correlation (FSC) = 0.143 criterion.

For the substrate-free inward-open state dataset, particle picking was performed as described above, and 2,807,898 particles were initially picked from 3474 micrographs. Particles were sorted as described above and 380,387 selected particles were subjected to 3D refinement, resulting in a 3.12 Å map. Several rounds of per-particle CTF refinement, Bayesian polishing, micelle subtraction and 3D classification without alignment were performed, and 117,385 particles were subjected to the final 3D refinement with C2 symmetry, which yielded the map with a global resolution of 2.60 Å.

For the intermediate and substrate-free occluded states dataset, particle picking was performed as described above and 3,699,304 particles were initially picked from 5,907 micrographs. After several

rounds of 2D classification, 3D classification resulted in two resolved classes, one with a top view resembling the inward-open state and the other with a distinct top view. For the 336,039 particles of the former class, Bayesian polishing, per-particle CTF refinement, micelle subtraction and 3D classification without alignment were performed, and 149,304 particles were subjected to final 3D refinement with C2 symmetry using SIDESPLITTER, which yielded the map with a global resolution of 3.47 Å. For the 448,031 particles of the latter class, several rounds of per-particle CTF refinement, Bayesian polishing, micelle subtraction, and 3D classification without alignment were performed, and 116,307 particles were subjected to the final 3D refinement with C2 symmetry using SIDESPLITTER, which yielded the map with a global resolution of 2.85 Å.

### Model building and refinement

The initial structural model of hSVCT1 was downloaded from the AlphaFold[54,55] Protein Structure Database (https://alphafold.ebi.ac.uk/). After fitting the predicted model into the substrate-bound inward-open state map by using MOLREP[56], the model was manually adjusted using COOT[57] and automatically refined by the phenix.real_space_refine program[58,59]. The models of the substrate-free inward-open state and the occluded state were built by using the model of the substrate-bound inward-open state as the initial model. The model of the intermediate state was built by using the model of the substrate-free inward-open state as the initial model. Finally, the models were refined using Servalcat[60]. The 3DFSC sphericity[61] was calculated by the Remote 3DFSC Processing Server (https://3dfsc.salk.edu). The statistics of the 3D reconstruction and model refinement are summarized in Supplementary Table 1. The hSVCT1 structure in the substrate-bound occluded state used in the molecular morphing animation was homology-modeled with the SWISS-MODEL[62] web server (https://swissmodel.expasy.org), based on the UraA crystal structure (PDB ID: 5XLS). All molecular graphics figures were prepared using PyMOL (Schrödinger), CueMol2 (http://www.cuemol.org), and UCSF ChimeraX[63].

### Substrate uptake assay

HEK293T cells were cultured in DMEM medium supplemented with 10% (v/v) FBS, at 37 °C with 5% $CO_2$. Cells were seeded in 12-well culture plates (Thermo Fisher Scientific) at a density of $4 \times 10^5$ cells/mL and transiently transfected with 500 ng of pEG BacMam vectors containing the hSVCT1 wild-type or mutant coding region with a C-terminal GFP tag, using the Lipofectamine 3000 transfection reagent. Cells were cultured for 48 h at 37 °C with 5% $CO_2$. For mock samples, HEK293T cells were cultured using same procedure, and no transfection was performed. For the time course assay, the culture medium was replaced with 1 mL of substrate-containing buffer (20 mM HEPES, pH 7.0, 150 mM NaCl, 5 mM KCl, 1 mM $MgCl_2$, 1 mM $CaCl_2$, 500 μM sodium ascorbate) and incubated for 1, 20, 40, and 60 min. For the measurement of transport kinetics, the culture medium was replaced with 1 mL of substrate-containing buffer with different sodium ascorbate concentrations (0, 10, 20, 50, 100, 200, and 500 μM) and incubated for 30 min. For the substrate uptake comparison of mutants, the culture medium was replaced with 1 mL of 500 μM substrate-containing buffer, and incubated for 30 min. Cells were harvested, washed twice with wash buffer (20 mM HEPES, pH 7.0, 150 mM NaCl, 5 mM KCl, 1 mM $MgCl_2$, 1 mM $CaCl_2$), and solubilized with 1% LMNG and 0.1% CHS for 1 h at 4 °C. Insoluble materials in the cell lysates were removed by ultracentrifugation at $108,000 \times g$ for 20 min at 4 °C, and the amounts of substrate in the supernatants were determined with an OxiSelect Ascorbic Acid Assay Kit (Cell Biolabs, STA-860). The absorbance at 595 nm was measured using a SpectraMax iD3 plate reader (Molecular Devices) with clear half-well 96-well plates (Greiner). All data analyses and curve-fitting to the Michaelis-Menten equation were performed using GraphPad Prism 9.5.1. To assess the ratio of dimer

formation for each mutant, 50 μL portions of the supernatants were subjected to GFP fluorescence-detection size-exclusion chromatography on a Superdex 200 Increase 10/300 GL column or an ENrich size-exclusion chromatography 650 10 × 300 column (Bio-Rad), and the curves were compared to that of wild-type.

### Confocal fluorescence microscopy imaging

HEK293T cells were cultured in DMEM medium supplemented with 10% (v/v) FBS, at 37 °C with 5% $CO_2$. Cells were seeded in 35 mm glass bottom dishes (MatTek) coated with Matrigel (Corning) at a density of $4 \times 10^5$ cells/mL and transiently transfected with 1 μg of pEG BacMam vectors containing the hSVCT1 wild-type or mutant coding region with a C-terminal GFP tag, using the Lipofectamine 3000 transfection reagent, and cultured for 48 h at 37 °C with 5% $CO_2$. Cells were cultured for 48 h at 37 °C with 5% $CO_2$, and then the culture medium was replaced with Leibovitz's L-15 medium without phenol red (Gibco) supplemented with 10% FBS. DNA was stained with Hoechst 33342 (Dojindo). Images were captured by an LSM 880 confocal laser scanning microscope (Zeiss) and analyzed using the ImageJ2 software (https://imagej.net/software/imagej2/).

### Reporting summary

Further information on research design is available in the Nature Portfolio Reporting Summary linked to this article.

## Data availability

Cryo-EM density maps and atomic coordinates of the structures presented in this manuscript have been deposited in the Electron Microscopy Data Bank (EMDB) and the Protein Data Bank (PDB) under the accession codes EMD-36201 and 8JEW (substrate-bound inward-open state), EMD-36204 and 8JEZ (substrate-free inward-open state), EMD-36205 and 8JF0 (intermediate state), and EMD-36206 and 8JF1 (substrate-free occluded state), respectively. Raw images have been deposited in the Electron Microscopy Public Image Archive under the accession code EMPIAR-12159. Source data are provided with this paper.

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

## Acknowledgements
We thank K. Ogomori and C. Harada for technical assistance, and K. Yamashita for assistance with model building. This work was supported by JSPS KAKENHI Grant Numbers JP21H05037 (O.N.), JP20K15754, JP22K15072 and JP24K01961 (T.K.), JP19H05794 and JP19H05795 (Y.O. and S.E.); JST CREST Grant Number JPMJCR20E2 (O.N and Y.O.), PRESTO Grant Number JPMJPR22E4 (T.K.) and Moonshot R&D Grant Number JPMJMS2025-14 (Y.O.); Research Support Project for Life Science and Drug Discovery (Basis for Supporting Innovative Drug Discovery and Life Science Research (BINDS)) from AMED under Grant Numbers JP23ama121002 (support number 3272) and JP23ama121012 (O.N.).

## Author contributions
T.A.K. designed and performed all experiments with assistance from H.S.; F.K.S and T.K. assisted with the cryo-EM data acquisition and single particle analysis. T.A.K. and T.K. performed the model building and structural refinement. Y.I. assisted with the functional analysis. S.E. and Y.O. assisted with the confocal microscopy analysis. H.S. conceived the project. T.A.K. and T.K. prepared the manuscript with input from all authors. T.K. and O.N. supervised the project.

## Competing interests
O.N. is a co-founder and scientific advisor for Curreio. All other authors declare no competing interests.
