## [Peer Review File · Nature Communications]

Dimeric transport mechanism of human vitamin C transporter SVCT1Reviewer #1 (Remarks to the Author):

Review

Novel dimeric transport mechanism of human vitamin C transporter SVCT1

Takaaki Kobayashi, Hiroto Shimada, Fumiya K. Sano, Tsukasa Kusakizako & Osamu Nureki

Outline

The present manuscript presents distinct structures of hSVCT1, a human Na/ascorbate symporter, obtained by cryo-EM. hSVCT1 is a member of the well-studied family of Nucleobase-Ascorbate Transporters (NATs), which has been extensively studied at the genetic and functional level in fungi and bacteria. Homologues of hSVCT1 exist in all domains of life and the structure of three of them has been resolved by crystallography or cryoEM. These are the *Escherichia coli* UraA uracil/H⁺ symporter, the *Aspergillus nidulans* UapA uric acid-xanthine/H⁺ symporter, and recently the mouse mSVCT1 ascorbate/Na⁺ symporter.

Structures described in the present manuscript were obtained in lipid (LMNG-CHS) - solubilized conditions, which favor dimerization, in the absence or presence of ascorbate, added during protein purification. In the presence of standard Na-supplemented media and purification buffers, two inward-open structures were obtained at high resolution (2.5-2.6 Å), either as substrate-loaded or substrate-free proteins. A third structure called intermediate is also reported at 3.6 Å, but this not discussed much in the text. The high resolution two inward-open structures are practical identical, except from the substrate binding site which is either loaded with ascorbate and two Na ion or 'empty'. In other words, the presence of substrates did not have a major effect on the inward-open structures obtained.

The inward-open structures reported are very similar to the previously published inward-open structures of homologues that have been structurally resolved, namely UraA, UapA or mSVCT1. On the inward-open structure of hSVCT1 the authors 'mapped' the residues interacting with ascorbate and the two Na ions, Na1 and Na2. Na1 neutralizes the negative charge of two acidic residues in the binding site (Glu334 and Asp338 in TM8) and thus enables binding of ascorbate, while Na2 seems to basically neutralize a negative charge on the substrate, via binding to Ser110 and Phe112 in TM3. This is very similar to what has been found for the binding of ascorbate and Na1 and Na2 in the homologous mouse mSVCT1 (Wang et al. *J. Nat Commun.* 2023), and also similar to what has been proposed for the binding of purines in UapA through neutralization by H⁺ of the analogous acidic residues in the substrate binding site (Diallinas, *Trends Biochem Sci.* 2021). A minimal mutational analysis is presented in the manuscript to provide some evidence for the functional importance of the residues proposed to act in ascorbate and Na binding.

Interestingly, a different hSVCT1 structure described in the manuscript, resolved at 2.85 Å, seems to have a very distinct topology from the inward-open structures. It shows an occluded topology, in an apo state, where there is a dramatic rearrangement of the dimer interface, so that several helices of the two monomers are now exposed and seem to interact with multiple LMNG and CHS lipids (e.g. TMs 6 and 13 are outwardly rotated). According to figure 4b one protomer is shifted compared to the inward-open dimer by ~20 Angstroms. In this case TM13 of one protomer is located between TM6 and TM7 of the other protomer while in the inward-open dimeric form TMS13 of one protomer is located between TMS12 and TMS6 of the other. Overall, this 'new' hSVCT1 topology is not forming a tight dimer, as the empty space between TM6 and TM12 is filled with lipids and surfactants. In normal dimers this space is occupied by TM13 of the other protomer. Curiously, this substrate-free occluded structure was obtained only when Na was replaced by K in all protein purification buffers. Noticeably also, no K or potassium ascorbate density was found in this structure. In other words, this structure was obtained merely due to the absence of Na ions from the purification and reconstitution steps. Such a structure, where the dimerization domains move apart and allow lipids to invade the space between them, has not been reported before not only for NAT homologues, but also for any of the 62 known and structurally similar elevator-type transporters (Trebesch & Tajkhorshid, 2023, *bioRxiv preprint* doi: <https://doi.org/10.1101/2023.06.14.544989>).

The authors propose that the substrate-free occluded new structure represents an intermediate conformational state during transition from the inward-open state to the substrate-free occluded state and outward-open conformations. Their statement was supported by a notable upward shift of elevator helices (e.g. TM10) towards the extracellular side, compared to the inward-open state.

To further support the functional importance of this new topology as a transient conformational intermediate in the hSVCT1 transport cycle, the authors mutated residues that are shown to form key interactions in "holding" the distinct dimer topologies in the inward-open and occluded states. Their results were used to propose that both dimer formations are true prerequisites for transport function.

Major criticism of the work reported

The manuscript basically describes two hSVCT1 structures obtained by cryoEM with the transporter protein purified, solubilized and reconstituted in lipids that support dimerization, in their case LMNG-CHS. The first structure is very similar to previously reported NAT structures, and in particular with the inward-open apo and substrate-bound states of mouse mSVCT1, determined in nanodiscs and lipids (POPC: POPG: POPE). The present inward-open structures of hSVCT1 are of higher resolution (2.5-2.6 versus 3.3-3.5), allowing somewhat better description of local details of substrate and Na ion binding, and defining the role of water molecules. However, overall, this new hSVCT1 structure does not provide any significant novel information on how SVCTs bind and transport ascorbate or how NAT function.

The potential novelty of this manuscript comes from the second very distinct hSVCT1 structure reported at 2.85 Å as a subclass of structures obtained in the absence of Na buffers. The new structure shows a dramatic loosening of dimerization due to rearrangement of one monomer relative to the other. This new topology of hSVCT1 structure shows two occluded monomers which are loosely connected via a several lipids (LMNG, CHS or other lipids coming from their preparation).

I am afraid I find very hard to be convinced that this structure, which no doubt was obtained under the author's experimental conditions, exists as a physiologically relevant intermediate state of transport catalysis *in vivo*, as the authors speculate. It will be extremely challenging thermodynamically to expose transiently and dynamically transmembrane helices of the dimerization domain in lipids, and alternate from a tight to loosened dimer. How can one explain the movement of the lipids in the conformational change seen between the resolved structures? From where would the lipids flow in the dimerization interphases? How do they move out for the next translocation cycle? Could this be an indication that the resolved occluded state is forced by conditions used to purify and reconstitute hSVCT1, and is thus an experimental artifact? I would suppose that the structure obtained might be a snapshot related to dimerization process. It is definitely not related to substrate transport. Unfortunately Molecular Dynamics are not capable to model this kind of large motions, and thus this structure does not help to dissect the details of transport catalysis in NAT transporters.

Furthermore, the fact that the key experiential modification that led to the novel occluded structure is the absence of Na for purification buffers does not seem sound or physiologically relevant. In NATs and most Na symporters sodium ions bind to specific residues of the substrate binding site to modify their electrostatic character and architecture, and thus enable the binding of physiological substrates (in this case ascorbate). To my knowledge, I do not know any cation symporter, at least among elevator transporters, that its topology is as dramatically modified as hSVCT1, simply by the mere absence of Na cations.

Given that this new structure is so provocative, the authors will need to provide evidence that this topology is part of the transport cycle and exists *in vivo*. They tried to do so through a very minimal mutational analysis of two residues proposed to interact stabilize this transient dimeric structure, namely N453 in TM13 and N230 in TM6. Their results are minimal and not convincing. First, they do not have any cellular studies to show whether the single and double mutations made (N230A, N453A and N230A/N453A) affect folding and trafficking of the transporter to the plasma membrane (this is a general comment for all the mutations reported in this work). Second, they do not present any rigorous kinetic analysis of hSVCT1 mutants. Ideally they should have used radiolabeled ascorbic acid and determine K_m and V_m values for ascorbate transport. Instead they used a biochemical kit that only gives a basic idea whether there is some function of the transporter. They determine indirectly whether these mutations affect dimerization by using FSEC,

but even so, their results do not support that interaction of N230 in one monomer with N453 in the other monomer is essential for dimerization in the new occluded structure. For example, notice that N230A does not affect dimerization, while N453 led to a mixture of dimers and monomers and the double mutation led to mostly monomers. If these two residues interact and affect dimerization, all three mutants should lead to equally reduced dimerization. As they are, these results show an additive detrimental effect of these mutations in hSVC1 function, probably by affecting its proper folding and function. Last but not least, N453 and N230 are only partially conserved in mammalian homologues, being replaced by other residues, often hydrophobic, in UraA or UapA. Given that NATs are not only structurally extremely similar but also homologous, I find difficult to reconcile the lack of evolutionary conservation in residues essential for transport catalysis, as all other residues previously shown to be essential for transport are indeed highly conserved in all NATs.

Finally, the map of another 'intermediate' state reported in Supplementary material, but not discussed in the text, is aggressively oversharpened, and thus I completely disagree with the reported resolution. If I had to guess, I'd say it is about 5-6 Å. After gaussian filtering so that only the tube of the helices is discernable, it becomes apparent that the model is clearly misbuilt. More specifically: No density exists for residues 382-394, even in really high thresholds. Residues 334-345 at the end of the helix seem to be modelled inaccurately. The density deviates medially from the model, up to a distance of 10 Å by residue 343. Nevertheless, they only mention this state just once in the text, and they do not base any of their mechanistic insight on this. I think the authors should not submit to the pdb this structure as it is, and redo the processing to get a healthy looking map, with a lower, and accurate, reported resolution

Specific points

Extended Data Fig. 1a: Repeat SEC experiments in the same conditions for the Na⁺ free preparation. Does the oligomeric behaviour of the protein change in a Na free state? We should expect so, because of the much reduced dimerization contact area in the conformation solved from this dataset.

Extended Data Fig. 2: The reported overall resolution of 2.45Å does not match the reported local resolution. According to 2.d, only the core reaches 2.4Å, while the periphery refines to a much lower resolution. Is the resolution factual?

Extended Data Fig. 3: Same as for Extended Data Fig. 2

Extended Data Figs. 4 and 9: Label lipids and detergents in all views.

Extended Data Fig 4d: Is the density modeled as POPC really a phospholipid, or could it be an LMNG? There exists some unexplained density which could very well belong to the second head group of an LMNG molecule. Show the density in higher threshold. The lipid molecule looks too distant to the protein density. Which are the interactions that would explain its resolution and its stable placement?

The model of the Occluded-Substrate free state contains two overlapping LMNG molecules, specifically the ones numbered AV0-603A and B, and one should be removed

In lines 236-238 the authors describe how important the formation of a dimer is for ascorbate transport. There is no detailed structural explanation for this finding and no experiments to support this statement. In comparison with UapA, which is shown experimentally to function only as a dimer, TM13 protrudes in the binding cavity of the second protomer and facilitates translocation of the substrates. Is there any hypothesis how dimers are functionally important for substrate translocation by hSVCT1?

Conclusion

I do not support publication of this work in Nat Com, unless it undergoes a major experimental revision, which will address specifically the physiological relevance of the new structure resolved. Below are some suggestions to this direction.

1. Perform an extensive mutational analysis of residues that seem to be involved in the two distinct dimeric states. For example, introduce salt bridges (i.e. pairs of E/D with K/R) to stabilize dimerization of the new occluded structures/ You might also use genetic data obtained with UapA to design mutations affecting dimerization (Kourkoulou et al. 2019) and lipid interactions (Pyle et al., 2018).
2. The functional analysis should include cellular studies using GFP-tagged versions of the transporter and rigorous kinetic analysis using radiolabeled ascorbate.
3. Given the successful use of bifluorescence assays in studying UapA (Martzoukou et al, 2015), use the same approach to address dimerization in the wt and mutant of hSVC1

Reviewer #2 (Remarks to the Author):

The manuscript by Kobayashi and Nureki et al titled "Novel dimeric transport mechanism of human vitamin C transporter SVCT1" reports cryoEM structures of human SVCT1 in substrate-bound and substrate-free states at resolutions of around 2.5 Angstrom. The main discoveries are 1) dimeric assembly of SVCT1, 2) visualization of the vitamin C binding site, 3) visualization of two Na ion binding sites, 4) structural changes at the vicinity of the substrate binding site that affect substrate accessibility, and 5) structural changes at the dimer interface that may be required for substrate translocation. Although the structure of the highly homologous mouse SVCT1 was published a few months ago, the current study provides significant new information on substrate conformation and its coordination due to the higher resolutions and the new apo structure reveals novel structural changes which may be relevant to the transport mechanism.

I have several comments that are more or less related to establishing more rigorous connections between novel features observed in the structures and their presumed or hypothesized functional relevance. Would be great if the authors could address these comments with new experiments but these comments should not be taken as demand for more experiments.

Measurement of Vitamin C transport by SVCT1 would be more precise if the authors could use a proteoliposome uptake assay. Radio-labelled Vit C is commercially available. The current assay is indirect, and the results could be further complicated by factors that are difficult to control or keep consistent.

It would also be good to have a real binding assay for vitamin C and for Na ions as well. ITC would be suitable, or scintillation proximity assay would be good too.

The assignment of the Na₂ site is very interesting because it is different from previous reports of Na ion binding sites. Because it has not been so common to see a Na ion stabilized by an aromatic ring, it would be more rigorous if the authors can sustain their claims with a Na ion binding assay and mutations.

The observed structural changes at the dimer interface is novel, and this is an aspect that has largely been ignored in the commonly invoked alternating access model for dimeric transporters. This is very exciting discovery, and it would catch more attention of the field if the authors could apply crosslinking to pinpoint interactions whose disruption or formation are required for substrate transport or binding.

Reviewer #3 (Remarks to the Author):

I have read the manuscript "Novel dimeric transport mechanism of human vitamin C transporter

SVCT1" by Nureki and coworkers with great interest. It reports several cryo-electron microscopy (cryoEM) structures of human vitamin C transporter SVCT1 in different conformational states. The structures provide insights into conformational changes during substrate transport and identified critical residues participating in the substrate binding and transport. The work provides valuable insights into the conformational changes and critical residues involved in substrate binding and transport of SVCT1. However, I have a few major and minor comments that I believe will enhance the clarity and impact of your findings.

Major Comments

1. My main concern is regarding the categorial processing using C2 symmetry. It would be crucial to attempt a reconstruction in C1 symmetry, and ideally perform symmetry expansion procedures, to investigate potential asymmetric states of the transporter. If there are conformational differences specific to each protomer, please speculate on the implications of these differences for vitamin transport.
2. In the FSEC experiments, the presence of CHS with LMNG resulted in a shift towards dimer formation compared to the absence of CHS. In the cryoEM structures, endogenous cholesterol molecules are modeled at the interface between the gate and core domains, while CHS molecules are modeled at the dimer interface. On what basis were densities assigned to either cholesterol or CHS, and how confident are the authors in these assignments. Additionally, please speculate on the potential roles of cholesterol in dimer formation and the conformational dynamics between the core and gate domains?
3. The substrate-binding pocket of SVCT1 is located at the end of a cleft between the gate and core domains, supported by an antiparallel β -strand sandwich. Please elaborate on the role of the β -strand sandwich in substrate binding and subsequent transport?
4. The authors describe that the sodium 1 (Na1) binding site in human SVCT1 is similar to the mouse SVCT1 whereas Na2 site differs. It would be helpful to include an extended data figure comparing the substrate-binding sites of mouse and human SVCT1. Does the electrostatic potential surface of the binding sites offer any insights into differences in Na2 binding site? Also, does the mouse SVCT1 also display non-selectivity towards vitamin C and its isomers?
5. Through structural and functional studies, the authors showed that the dimer interface is important for substrate transport. Is there any cooperativity or influence of one protomer's substrate binding status on the second protomer's function?
6. Based on the cryoEM structures of SVCT1 in different states, the authors propose a transport mechanism combining elevator and rotational motions-driven conformational transitions for substrate binding, transport, and release. It is an exciting claim for the transport of vitamin C through SVCT1. Do the authors have any additional theoretical or experimental evidence to support their transport mechanism?
7. Judging from the 2D classes it appears that the datasets show strong preferred orientation (mostly top views). Orientational distribution is not reported anywhere. Please update the Extended Data Figures with a representation and include a plot for 3DFSC figure of merit.
8. Figures 1 and 3 should be merged into one figure to have a more direct comparison between the structures. The same is true for Figures 5 and 6, which should be merged.

Minor Comments

1. Line 181: The line reads, "substrate-binding pocket may be be too small to accommodate...". The word "be" is repeated twice.
2. Figure 1 legend, lines 415-416: "The core and gate domains of a protomer are colored light blue and deep blue, respectively." I suggest referring to the light blue color as "cyan" for clarity.
3. In relation to minor comment 2, the Figure 5 legend mentions that the inward-open hSVCT1 structure is colored blue. I suggest referring to the blue color as "cyan" for clarity.

Reviewer #1:

OUTLINE

The present manuscript presents distinct structures of hSVCT1, a human Na/ascorbate symporter, obtained by cryo-EM. hSVCT1 is a member of the well-studied family of Nucleobase-Ascorbate Transporters (NATs), which has been extensively studied at the genetic and functional level in fungi and bacteria. Homologues of hSVCT1 exist in all domains of life and the structure of three of them has been resolved by crystallography or cryoEM. These are the Escherichia coli UraA uracil/H⁺ symporter, the Aspergillus nidulans UapA uric acid-xanthine/H⁺ symporter, and recently the mouse mSVCT1 ascorbate/Na⁺ symporter. Structures described in the present manuscript were obtained in lipid (LMNG-CHS) - solubilized conditions, which favor dimerization, in the absence or presence of ascorbate, added during protein purification. In the presence of standard Na-supplemented media and purification buffers, two inward-open structures were obtained at high resolution (2.5-2.6 Å), either as substrate-loaded or substrate-free proteins. A third structure called intermediate is also reported at 3.6 Å, but this not discussed much in the text. The high resolution two inward-open structures are practical identical, except from the substrate binding site which is either loaded with ascorbate and two Na ion or 'empty'. In other words, the presence of substrates did not have a major effect on the inward-open structures obtained.

The inward-open structures reported are very similar to the previously published inward-open structures of homologues that have been structurally resolved, namely UraA, UapA or mSVCT1. On the inward-open structure of hSVCT1 the authors 'mapped' the residues interacting with ascorbate and the two Na ions, Na1 and Na2. Na1 neutralizes the negative charge of two acidic residues in the binding site (Glu334 and Asp338 in TM8) and thus enables binding of ascorbate, while Na2 seems to basically neutralize a negative charge on the substrate, via binding to Ser110 and Phe112 in TM3. This is very similar to what has been found for the binding of ascorbate and Na1 and Na2 in the homologous mouse mSVCT1 (Wang et al. J. Nat Commun. 2023), and also similar to what has been proposed for the binding of purines in UapA through neutralization by H⁺ of the analogous acidic residues in the substrate binding site (Diallinas, Trends Biochem Sci. 2021). A minimal mutational analysis is presented in the manuscript to provide some evidence for the functional importance of the residues proposed to act in ascorbate and Na binding.

Interestingly, a different hSVCT1 structure described in the manuscript, resolved at 2.85 Å, seems to have a very distinct topology from the inward-open structures. It shows an occluded topology, in an apo state, where there is a dramatic rearrangement of the dimer interface, so

that several helices of the two monomers are now exposed and seem to interact with multiple LMNG and CHS lipids (e.g. TMs 6 and 13 are outwardly rotated). According to figure 4b one protomer is shifted compared to the inward-open dimer by ~20 Angstroms. In this case TM13 of one protomer is located between TM6 and TM7 of the other protomer while in the inward-open dimeric form TM13 of one protomer is located between TM12 and TM6 of the other. Overall, this ‘new’ hSVCT1 topology is not forming a tight dimer, as the empty space between TM6 and TM12 is filled with lipids and surfactants. In normal dimers this space is occupied by TM13 of the other protomer.

Curiously, this substrate-free occluded structure was obtained only when Na was replaced by K in all protein purification buffers. Noticeably also, no K or potassium ascorbate density was found in this structure. In other words, this structure was obtained merely due to the absence of Na ions from the purification and reconstitution steps. Such a structure, where the dimerization domains move apart and allow lipids to invade the space between them, has not been reported before not only for NAT homologues, but also for any of the 62 known and structurally similar elevator-type transporters (Trebesch & Tajkhorshid, 2023, bioRxiv preprint doi: <https://doi.org/10.1101/2023.06.14.544989>).

The authors propose that the substrate-free occluded new structure represents an intermediate conformational state during transition from the inward-open state to the substrate-free occluded state and outward-open conformations. Their statement was supported by a notable upward shift of elevator helices (e.g. TM10) towards the extracellular side, compared to the inward-open state. To further support the functional importance of this new topology as a transient conformational intermediate in the hSVCT1 transport cycle, the authors mutated residues that are shown to form key interactions in ‘holding’ the distinct dimer topologies in the inward-open and occluded states. Their results were used to propose that both dimer formations are true prerequisites for transport function.

MAJOR CRITICISM OF THE WORK REPORTED

The manuscript basically describes two hSVCT1 structures obtained by cryoEM with the transporter protein purified, solubilized and reconstituted in lipids that support dimerization, in their case LMNG-CHS. The first structure is very similar to previously reported NAT structures, and in particular with the inward-open apo and substrate-bound states of mouse mSVCT1, determined in nanodiscs and lipids (POPC: POPG: POPE). The present inward-open structures of hSVCT1 are of higher resolution (2.5-2.6 versus 3.3-3.5), allowing somewhat better description of local details of substrate and Na ion binding, and defining the role of water

molecules. However, overall, this new hSVCT1 structure does not provide any significant novel information on how SVCTs bind and transport ascorbate or how NAT function.

We thank the reviewer for the comments. As you mentioned, the overall conformation of our human SVCT1 inward-open structures in LMNG-CHS micelles is similar to the previously reported mouse SVCT1 structures. However, we certainly believe that the improvement in resolution from 3.3 Å to 2.5 Å is a significant advancement in the field of structural biology. The higher resolution enables a more detailed description of the substrate binding mode. The density of the bound ascorbate in our structure is much clearer than that in the mouse structure, and we could accurately determine the orientation of the substrate in this state (**Fig. L1a, b**). Moreover, the environment surrounding the substrate is well resolved, allowing us to model the responsible residue and two Na⁺ ions in more precise positions, consistent with the charge distribution of the ascorbate molecule. Additionally, the cryo-EM map obtained under the Na⁺-free condition shows the density corresponding to a water molecule interacting with the amide of A113 (**Fig. L1c**), supporting our assignment of a water molecule instead of Na⁺ (Na2 in the previous mouse SVCT1 structure). These clarifications, in addition to the structure of a new conformation, lead us to propose a more confident substrate recognition mechanism, and we consider it to be a substantial update from the previous report.

Fig. L1 | Comparison of human and mouse SVCT1 density maps around the bound substrate. (a) Human SVCT1 density map (EMD-36201) and substrate-bound inward-open model (PDB 8JEW). **(b)** Mouse SVCT1 density map (EMD-34094) and substrate-bound inward-open model (PDB 7YTW). Both F_o maps are shown at the same contour level. **(c)** Water molecules in the substrate-free occluded state obtained in the Na^+ -free condition. Green mesh represents F_o-F_c omit map.

The potential novelty of this manuscript comes from the second very distinct hSVCT1 structure reported at 2.85 Å as a subclass of structures obtained in the absence of Na buffers. The new structure shows a dramatic loosening of dimerization due to rearrangement of one monomer relative to the other. This new topology of hSVCT1 structure shows two occluded monomers which are loosely connected via a several lipids (LMNG, CHS or other lipids coming from their preparation).

I am afraid I find very hard to be convinced that this structure, which no doubt was obtained under the author's experimental conditions, exists as a physiologically relevant intermediate state of transport catalysis in vivo, as the authors speculate. It will be extremely challenging thermodynamically to expose transiently and dynamically transmembrane helices of the dimerization domain in lipids, and alternate from a tight to loosened dimer. How can one explain the movement of the lipids in the conformational change seen between the resolved structures? From where would the lipids flow in the dimerization interphases? How do they move out for the next translocation cycle? Could this be an indication that the resolved occluded state is forced by conditions used to purify and reconstitute hSVCT1, and is thus an experimental artifact? I would suppose that the structure obtained might be a snapshot related to dimerization process. It is definitely not related to substrate transport. Unfortunately Molecular Dynamics are not capable to model this kind of large motions, and thus this structure does not help to dissect the details of transport catalysis in NAT transporters.

Thank you for the insightful comments. We investigated the relationship between solubilization conditions and monomer-dimer formation of hSVCT1 in our study, and found that a simple alteration of detergents (LMNG to DDM) hinders dimer formation and results in larger monomer populations in micelles (**Fig. L2**). We believe this is evidence of the relatively flexible dimer interface formed under both buffer conditions (**Fig. L2a, b**). Thus, although the dimer interface of the inward-open state appears to be tightly packed, conformational changes of the dimer interface are conceivable through interactions between protein domains and lipids. We also observed lipids bound at the protein surface near the dimer interface, which led us to speculate that membrane lipids are incorporated within the interface via conformational changes.

As the reviewer pointed out, the possibility of the novel dimeric state being an experimental artifact, due to the buffer conditions of purification and micelle reconstitution, cannot be completely ruled out. To address this, we verified if the re-dimerization of hSVCT1 monomers occurs in Na⁺-free conditions. We purified monomeric hSVCT1 with DDM in the absence of Na⁺, and then exchanged the detergent into LMNG to reform dimers. However, we found that the extracted monomeric hSVCT1 did not form dimers again in the solutions (**Fig. L2c**), thus re-dimerization of hSVCT1 monomers seems unlikely to occur. This supports our belief that the conformational change into the novel occluded structure is caused by the exposure to Na⁺-free buffer conditions in the plasma membrane before extraction, and those hSVCT1 molecules are extracted from the membrane as dimers. The emergence of two conformational classes

corresponding to both inward and occluded states during the single particle analysis of the Na⁺-free dataset also supports this structural equilibrium.

Fig. L2 | Oligomeric behavior of hSVCT1 under different solubilization conditions. (a) FSEC chromatograms of hSVCT1 with different detergents under Na⁺-containing conditions. (b) FSEC chromatograms of hSVCT1 with different detergents under Na⁺-free conditions (i.e., K⁺ ions were used instead of Na⁺ ions). (c) Detergent substitution of DDM-solubilized monomers to LMNG under Na⁺-free conditions (i.e., K⁺ ions were used instead of Na⁺ ions), represented by FSEC chromatograms. GFP fluorescence was detected in all chromatograms.

Furthermore, the fact that the key experiential modification that led to the novel occluded structure is the absence of Na for purification buffers does not seem sound or physiologically relevant. In NATs and most Na symporters sodium ions bind to specific residues of the substrate binding site to modify their electrostatic character and architecture, and thus enable the binding of physiological substrates (in this case ascorbate). To my knowledge, I do not know any cation symporter, at least among elevator transporters, that its topology is as dramatically modified as hSVCT1, simply by the mere absence of Na cations.

Thank you for the insightful comments. In a previous study of Na⁺ symporters, the Na⁺-free conditions were used to capture the apo state of the Na⁺-symporter VcINDY, indicating the

local structural flexibility (Nat. Commun. 13, 2644, 2022). We also applied the Na⁺-free conditions to obtain different states since the absence of Na⁺ may trigger a multi-step conformational change, given the crucial role of Na⁺ in the transport mechanism of hSVCT1. Consequently, we obtained two distinct states from the same Na⁺-free dataset: the inward-open-like intermediate state with local structural flexibility, and surprisingly, the substrate-free occluded state with a drastic rearrangement of the dimer interface. The latter unprecedented state is considered to capture a state after the release of the substrate and Na⁺ ions in the inward-open state and before the next ones are imported in the outward-open state, as cation symporters (importers) are known to adopt the apo state during transitions from the inward-facing to outward-facing states. It would be difficult to replicate this state in other NAT members because many of them are H⁺-coupled symporters, and excluding protons from the purification environment is impossible.

Given that this new structure is so provocative, the authors will need to provide evidence that this topology is part of the transport cycle and exists in vivo. They tried to do so through a very minimal mutational analysis of two residues proposed to interact stabilize this transient dimeric structure, namely N453 in TM13 and N230 in TM6. Their results are minimal and not convincing. First, they do not have any cellular studies to show whether the single and double mutations made (N230A, N453A and N230A/N453A) affect folding and trafficking of the transporter to the plasma membrane (this is a general comment for all the mutations reported in this work).

Thank you for the helpful comments. As you pointed out, the mutational analysis was minimal, and the membrane expression of the mutants was not confirmed in the original manuscript. Therefore, we performed additional experiments to address these concerns.

Specifically, we designed several additional mutants of the residues at the dimer interface for each state, including hydrophobic interactions (F459A, F463A, F466A, F200W, W211A, L223W), with some of them also interacting with the lipids. We then assessed their membrane expression, dimer formation, and transport activities, and included the results in the revised manuscript (**Figs. L3, L4**, Fig. 5 and Supplementary Fig. 12).

First, we examined the membrane expression of wild-type hSVCT1 and its mutants by confocal microscopy, using the fused GFP fluorescence. The results, shown in **Fig. L3** and Supplementary Fig. 12, confirm the proper plasma membrane expression for the wild-type and each mutant.

GFP (hSVCT1)
Hoechst 33342 (DNA)

Fig. L3 | Plasma membrane expression of hSVCT1 mutants. Representative fluorescence images of HEK293T cells expressing C-terminally GFP-tagged hSVCT1 mutants. Nuclear DNA was stained with Hoechst 33342. The observations were repeated at least three times independently with similar results.

We next examined the dimer formation of each mutant by FSEC experiments (**Fig. L4** and **Fig. 5**). The results showed monodisperse peaks with similar widths to that of wild-type, and confirmed the structural integrity of each mutant (**Fig. L4a–e**). All mutants exhibited a decreased dimer population and increased monomer population. The transport assay for dimer interface mutants showed decreased transport activities (**Fig. L4e, f**), indicating that dimer formation plays a crucial role in the transport activity. We believe these additional results further support our hypothesis of the new conformational change and transport mechanism.

Fig. L4 | Dimer formation and transport activity of hSVCT1. (a–d) Close-up views of the residues responsible for dimer formation in the inward-open state (a, b) and substrate-free occluded state (c, d). (e) Evaluation of oligomeric populations of dimer interface mutants by FSEC, detected by GFP fluorescence. (f) Substrate uptake assay for dimer interface mutants. Values are mean of $n = 3$ biological replicates \pm s.e.m. compared relative to the wild-type uptake rate.

Second, they do not present any rigorous kinetic analysis of hSVCT1 mutants. Ideally they should have used radiolabeled ascorbic acid and determine K_m and V_m values for ascorbate transport. Instead they used a biochemical kit that only gives a basic idea whether there is some function of the transporter. They determine indirectly whether these mutations affect dimerization by using FSEC, but even so, their results do not support that interaction of N230 in one monomer with N453 in the other monomer is essential for dimerization in the new occluded structure. For example, notice that N230A does not affect dimerization, while N453 led to a mixture of dimers and monomers and the double mutation led to mostly monomers. If these two residues interact and affect dimerization, all three mutants should lead to equally reduced dimerization. As they are, these results show an additive detrimental effect of these mutations in hSVCT1 function, probably by affecting its proper folding and function.

Following the reviewer's suggestion, we planned to perform a liposome assay using radiolabeled ascorbate to measure the transport kinetics of the mutants. Before using radiolabeled ascorbate, we tried to confirm that hSVCT1 can be reconstituted into liposomes and retains the transport activity, by using a colorimetric assay kit. Although we successfully reconstituted hSVCT1 into liposomes with several lipid compositions, including the POPC:POPG:POPE mixture used in the mouse SVCT1 nanodiscs (**Fig. L5a–b**), no transport activity was observed among all conditions (**Fig. L5c**). The fact that hSVCT1 overexpressed on HEK293T cells is active in transport but becomes inactive after liposome reconstitution suggests that the complicated endogenous lipid composition of the plasma membrane is essential for the transport activity of hSVCT1, which is consistent with the speculation that lipid binding influences the conformational change.

Fig. L5 | Results of liposome assay trials. (a) FSEC chromatograms for hSVCT1 proteoliposomes. After the reconstituted proteoliposome solution was ultracentrifuged, the

supernatant (blue line, not containing liposomes) and detergent-solubilized precipitate (red line, containing liposomes) were subjected to size-exclusion chromatography to detect the existence of proteins. **(b)** SDS-PAGE of hSVCT1 proteoliposomes. The lanes represent the whole proteoliposome solution, the ultracentrifuged supernatant, and the precipitate, from left to right. **(c)** Transport kinetics assay for hSVCT1-reconstituted proteoliposomes. The dotted line represents the uptake curve obtained by the cellular assay as a reference.

Based on the above results, we concluded that the liposome assay is not suitable for hSVCT1, and we next tried using radiolabeled ascorbate with the cellular uptake assay. However, the transport activity of the radiolabeled isotope was noticeably detected even in non-transfected HEK293T cells, and the ascorbate uptake mediated by hSVCT1 was difficult to detect with this assay system (**Fig. L6**).

Fig. L6 | Scintillation counts of transported C-14 radioisotope. **(a)** hSVCT1-overexpressing or **(b)** non-transfected HEK293T cells were exposed to buffer containing ^{14}C -radiolabeled ascorbate. Cells were collected immediately or after 30 minutes, and the lysates were subjected to scintillation counting. This experimental procedure is based on the successfully conducted colorimetric cellular assay.

This may be due to the endogenous expression of other transporters, such as GLUTs, which transport the oxidized form of ascorbate. As ascorbate is prone to oxidation, oxidized radiolabeled ascorbate may be imported into cells via these transporters. The linearity of the plots also supports this hypothesis. The oxidized ascorbate is detected together with the reduced form in RI-based assays, making it difficult to isolate and specifically detect the reduced form of ascorbate transported by hSVCT1 using this system.

Therefore, we concluded that liposome- or RI-based systems cannot be applied to an hSVCT1 substrate transport assay, and decided to keep using the FRASC assay kit, which assesses the

reduction activity of ascorbic acid. Several reports, including the mouse SVCT1 paper, have demonstrated that this assay kit and its chemical principles ensure the specific and quantitative measurements of only the reduced form of ascorbic acid. Using this strategy, we successfully obtained a Michaelis-Menten plot for each hSVCT1 mutant, including the additional ones, and estimated their transport kinetics (Fig. L7 and Supplementary Fig. 9).

Fig. L7 | Transport kinetics analysis of hSVCT1 mutants. Values are mean of $n = 3$ biological replicates \pm s.e.m. Asterisk indicates that the estimated K_m value exceeds the upper

limit of the measurement range. All kinetics values and K_m curves were calculated using GraphPad Prism 9.5.1.

In response to the reviewer's comment, we also re-examined the oligomer formation by N230A- including single and double mutants (**Fig. L8**). In several repeated experiments, the typical results showed obvious double peaks corresponding to both monomeric and dimeric forms, confirming that the N230A mutation also leads to a mixture of monomers and dimers (**Fig. L8a**). In addition, the dimer peak of the N230A/N453A double mutant at the 10 mL elution volume is quite low compared to those of both the N230A and N453A single mutants (**Fig. L8a**). These results indicate that this combination of mutants effectively inhibits dimer formation, while not affecting the protein folding. The curves shown in the original manuscript indicated only a slight increase in the population of monomers (**Fig. L8b**). However, we found this to be the minor result, so we replaced the figure with the major example in the revised manuscript.

Fig. L8 | Re-examination of oligomeric populations for N230A-related mutants. (a) Typical curves obtained from the re-examination in the revised manuscript. **(b)** Previously shown curves in the original manuscript.

Last but not least, N453 and N230 are only partially conserved in mammalian homologues, being replaced by other residues, often hydrophobic, in UraA or UapA. Given that NATs are not only structurally extremely similar but also homologous, I find difficult to reconcile the lack of evolutionary conservation in residues essential for transport catalysis, as all other residues previously shown to be essential for transport are indeed highly conserved in all NATs.

As the reviewer pointed out, the mutated residues at the dimer interface are only conserved in mammalian homologs, and not in other bacterial and fungal homologs such as UraA or UapA.

Indeed, the highly conserved residues among all NAT members are neither oriented toward the other protomer nor involved in the dimer formation (**Fig. L9**). Moreover, most of the residues forming the dimer interface are hydrophobic and not forming tight hydrophilic interactions, as newly added dimer-interface mutants in the transport assay, except for N230, Q448, and N453. This suggests that the stable formation of the dimer interface may not be conserved in the NAT family, and instead the flexibility of the dimer interface is important.

Fig. L9 | Conserved residues in the gate domain of hSVCT1. a Sequence alignment of NAT members. Residues corresponding to transmembrane helices of the core domain, transmembrane helices of the gate domain, and the β -strand regions are indicated with cyan, blue, and orange

bars, respectively. The conserved residues among all four members are indicated as red. **b** Illustration of the highly conserved residue sidechains in the gate domain, showing that no sidechain is oriented toward the dimer interface.

Finally, the map of another ‘intermediate’ state reported in Supplementary material, but not discussed in the text, is aggressively oversharpened, and thus I completely disagree with the reported resolution. If I had to guess, I’d say it is about 5-6 Å. After gaussian filtering so that only the tube of the helices is discernable, it becomes apparent that the model is clearly misbuilt. More specifically: No density exists for residues 382-394, even in really high thresholds. Residues 334-345 at the end of the helix seem to be modelled inaccurately. The density deviates medially from the model, up to a distance of 10 Å by residue 343. Nevertheless, they only mention this state just once in the text, and they do not base any of their mechanistically insight on this. I think the authors should not submit to the pdb this structure as it is, and redo the processing to get a healthy looking map, with a lower, and accurate, reported resolution.

Thank you for the helpful comment. According to the reviewer’s suggestion, we re-inspected our ‘intermediate’ state structure map and model. We re-calculated the appropriately weighted and sharpened map using the Servalcat pipeline (Acta Cryst. D77, 1282–1291, 2021), which is reasonable for an overall resolution of 3.5 Å. Based on this appropriately sharpened map, we found that residues 334–345 were incorrectly built and there was almost no density for residues 382–394. Therefore, we re-built the residues 334–345 to fit better into the map and removed the atomic models of residues 382–394, improving the quality of the intermediate state structure (**Fig. L10** and Supplementary Fig. 11). These changes highlight the differences between the intermediate and inward-open states of the intracellular region, despite their overall similarity. Even though the intermediate state only represents the mobility of the intracellular region, it is worth noting that the inward-open-like intermediate state and the substrate-free occluded state structures emerged from the same cryo-EM dataset (the Na⁺-free condition), suggesting the conformational equilibrium between two distinct states, as described in the manuscript.

Fig. L10 | Structure of hSVCT1 in the putative intermediate state between the inward-open and substrate-free occluded states. a Cryo-EM map of the hSVCT1 homodimer in the intermediate state, with densities corresponding to each protomer colored green and grey, respectively. **b** Overall structure of hSVCT1 in the intermediate state. The core and gate domains of a protomer are colored neon green and dark green, respectively.

SPECIFIC POINTS

Extended Data Fig. 1a: Repeat SEC experiments in the same conditions for the Na⁺ free preparation. Does the oligomeric behaviour of the protein change in a Na free state? We should expect so, because of the much reduced dimerization contact area in the conformation solved from this dataset.

Following the reviewer's comment, we repeated the FSEC experiments to examine the oligomeric behavior of hSVCT1 with different detergents in Na⁺-free conditions. The results demonstrated similar oligomer formation of hSVCT1 in Na⁺-free conditions compared to Na⁺-containing conditions (**Fig. L2a, b**). This suggests that even with reduced contact area, the substrate-free occluded conformation still forms a dimer, which may be facilitated by lipids or detergent molecules.

Fig. L2 | Oligomeric behavior of hSVCT1 under different solubilization conditions. (a) FSEC chromatograms of hSVCT1 with different detergents under Na⁺-containing conditions. **(b)** FSEC chromatograms of hSVCT1 with different detergents under Na⁺-free conditions (i.e., K⁺ ions were used instead of Na⁺ ions).

Extended Data Fig. 2: The reported overall resolution of 2.45Å does not match the reported local resolution. According to 2.d, only the core reaches 2.4Å, while the periphery refines to a much lower resolution. Is the resolution factual?

Extended Data Fig. 3: Same as for Extended Data Fig. 2

We thank the reviewer for the helpful comments. After reviewing our data, we found that the local resolution range was limited to 2.4 Å resolution, due to the default window size of 25 Å used in the local resolution sampling in RELION. Therefore, we re-calculated the local resolution using a smaller window size of 8 Å in the substrate-bound inward-open map (**Fig. L11a**) and the substrate-free inward map (**Fig. L11b**). These resulted in the local resolution range from 2.0 Å to ~4 Å (**Fig. L11b**), which is reasonable for the overall resolutions. All local resolution calculations have been updated in the revised manuscript (Supplementary Figs. 2–4).

Fig. L11 | Local resolution of inward-open maps. Histogram of local resolution distribution and surface color representation for (a) substrate-bound inward-open map, and (b) substrate-free inward-open map.

Extended Data Figs. 4 and 9: Label lipids and detergents in all views.

Following the reviewer's suggestion, we have labeled lipids and detergents in the revised figures.

Extended Data Fig 4d: Is the density modeled as POPC really a phospholipid, or could it be an LMNG? There exists some unexplained density which could very well belong to the second head group of an LMNG molecule. Show the density in higher threshold. The lipid molecule looks too distant to the protein density. Which are the interactions that would explain its resolution and its stable placement?

We appreciate the reviewer for this insightful suggestion. As the reviewer pointed out, we noticed some unmodeled densities, probably corresponding to the second head group of an LMNG molecule. Following the reviewer's suggestion, we modeled an LMNG molecule instead of a phospholipid (**Fig. L12**). The acyl chain of the LMNG molecule interacts with the

aromatic sidechain of Y227 (**Fig. L12**). We have updated Supplementary Fig. 5d to show this interaction.

Fig. L12 | Density of the LMNG molecule and its interacting Y227 sidechain. $F_o - F_c$ omit map (green mesh) of the LMNG molecule and F_o map (blue mesh) of the Y227 sidechain.

The model of the Occluded-Substrate free state contains two overlapping LMNG molecules, specifically the ones numbered AV0-603A and B, and one should be removed.

Thank you for this comment. In the model building and refinement of all structures, we manually built the asymmetric unit model (i.e., protomer) and refined it by using the structure refinement program REFMAC5 via the Servalcat pipeline (Acta Cryst. D77, 1282–1291, 2021), which refines a model in the asymmetric unit with symmetry constraints, as commonly done in crystallography. For the substrate-free occluded state, there is an LMNG molecule (AV0-603) located at a C2 symmetry axis between protomers. Therefore, we modeled one entire molecule of LMNG with an occupancy of 0.5 (AV0-603A) in the asymmetric unit and refined it with C2 symmetry constraints, resulting in the symmetric one with an occupancy of 0.5 (AV0-603B) at the C2 symmetry axis. We believe this procedure and the resulting model, representing two slightly different conformations of LMNG, each with an occupancy of 0.5 at a 2-fold symmetry axis, is reasonable.

In lines 236-238 the authors describe how important the formation of a dimer is for ascorbate transport. There is no detailed structural explanation for this finding and no experiments to

support this statement. In comparison with UapA, which is shown experimentally to function only as a dimer, TM13 protrudes in the binding cavity of the second protomer and facilitates translocation of the substrates. Is there any hypothesis how dimers are functionally important for substrate translocation by hSVCT1?

Our hypothesis proposes that the structural change of the dimer interface is important for driving the transport cycle. Without dimer formation, there are no interactions between the gate domains, thus preventing their necessary conformational changes such as the tilting movement to close the transport pathway. This hypothesis is supported by the experimental evidence demonstrating the decreased transport activity of the dimer interface-disrupting mutant, which basically corresponds to the ratio of dimer formations.

CONCLUSION

I do not support publication of this work in Nat Com, unless it undergoes a major experimental revision, which will address specifically the physiological relevance of the new structure resolved. Below are some suggestions to this direction.

- 1. Perform an extensive mutational analysis of residues that seem to be involved in the two distinct dimeric states. For example, introduce salt bridges (i.e. pairs of E/D with K/R) to stabilize dimerization of the new occluded structures/ You might also use genetic data obtained with UapA to design mutations affecting dimerization (Kourkoulou et al. 2019) and lipid interactions (Pyle et al., 2018).*
- 2. The functional analysis should include cellular studies using GFP-tagged versions of the transporter and rigorous kinetic analysis using radiolabeled ascorbate.*
- 3. Given the successful use of bifluorescence assays in studying UapA (Martzoukou et al, 2015), use the same approach to address dimerization in the wt and mutant of hSVCT1*

Thank you for the helpful comments. According to the reviewer's comments, we have added an extensive mutational analysis of the residues potentially involved in dimer formation and conformational changes. We confirmed the membrane localization by confocal microscopy, and obtained the evidence of the presence of this novel occluded conformation by detailed kinetic analyses of these mutants. We apologize that we could not accomplish some of the proposed experiments as we found that the liposome-based transport analysis using radiolabeled ascorbate is basically impossible due to the intrinsic nature of this transporter as described above, while we spent much time struggling with the liposome and radioisotope trials.

For dimerization of the wild-type and mutant SVCT1, instead of bifluorescence assays, we have designed additional interface-disrupting mutants for each dimeric state and examined the relationship between dimer formation and transport activity (**Fig. L4**). These additional results also support our hypothesis of the new transport mechanism. Altogether, we believe that the additional information included in our revised manuscript is sufficient to explain the novel occluded structure and the conformational change mechanisms. We look forward to your positive consideration.

Fig. L4 | Dimer formation and transport activity of hSVCT1. (a–d) Close-up views of the residues responsible for dimer formation in the inward-open state (a, b) and substrate-free occluded state (c, d). (e) Evaluation of oligomeric populations of dimer interface mutants by FSEC, detected by GFP fluorescence. (f) Substrate uptake assay for dimer interface mutants. Values are mean of $n = 3$ biological replicates \pm s.e.m. compared relative to the wild-type uptake rate.

Reviewer #2:

The manuscript by Kobayashi and Nureki et al titled "Novel dimeric transport mechanism of human vitamin C transporter SVCT1" reports cryoEM structures of human SVCT1 in substrate-

bound and substrate-free states at resolutions of around 2.5 Angstrom. The main discoveries are 1) dimeric assembly of SVCT1, 2) visualization of the vitamin C binding site, 3) visualization of two Na ion binding sites, 4) structural changes at the vicinity of the substrate binding site that affect substrate accessibility, and 5) structural changes at the dimer interface that may be required for substrate translocation. Although the structure of the highly homologous mouse SVCT1 was published a few months ago, the current study provides significant new information on substrate conformation and its coordination due to the higher resolutions and the new apo structure reveals novel structural changes which may be relevant to the transport mechanism.

I have several comments that are more or less related to establishing more rigorous connections between novel features observed in the structures and their presumed or hypothesized functional relevance. Would be great if the authors could address these comments with new experiments but these comments should not be taken as demand for more experiments.

We thank the reviewer for the comments. We have addressed the comments raised by the reviewer as follows.

Measurement of Vitamin C transport by SVCT1 would be more precise if the authors could use a proteoliposome uptake assay. Radio-labelled Vit C is commercially available. The current assay is indirect, and the results could be further complicated by factors that are difficult to control or keep consistent. It would also be good to have a real binding assay for vitamin C and for Na ions as well. ITC would be suitable, or scintillation proximity assay would be good too.

Thank you for the insightful comment. According to the reviewer's suggestion, we planned to perform a liposome assay using radiolabeled ascorbate to measure the transport kinetics of the mutants. Before using the radiolabeled ascorbate, we tried to confirm that hSVCT1 can be reconstituted into liposomes and retain the transport activity, by using the colorimetric assay kit. Although we successfully reconstituted hSVCT1 into liposomes with several lipid compositions, including the POPC: POPG: POPE mixture used in the mouse SVCT1 nanodiscs (**Fig. L5a–b**), no transport activity was observed among all conditions (**Fig. L5c**). The fact that hSVCT1 overexpressed on HEK293T cells is active in transport but becomes inactive after liposome reconstitution suggests that the complicated endogenous lipid composition of the

plasma membrane is quite essential for the transport activity of hSVCT1, which is consistent with the speculation that lipid binding is important for conformational changes.

Fig. L5 | Results of liposome assay trials. (a) FSEC chromatograms for hSVCT1 proteoliposomes. After the reconstituted proteoliposome solution was ultracentrifuged, the supernatant (blue line, not containing liposomes) and detergent-solubilized precipitate (red line, containing liposomes) were subjected to size-exclusion chromatography to detect the existence of proteins. **(b)** SDS-PAGE of hSVCT1 proteoliposomes. The lanes represent the whole proteoliposome solution, the ultracentrifuged supernatant, and the precipitate, from left to right. **(c)** Transport kinetics assay for hSVCT1-reconstituted proteoliposomes. The dotted line represents the uptake curve obtained by the cellular assay as a reference.

Based on the above results, we concluded that the liposome assay is not suitable for hSVCT1, and we next tried using radiolabeled ascorbate with the cellular uptake assay. However, the transport activity of the radiolabeled isotope was noticeably detected even in non-transfected

HEK293T cells, and the ascorbate uptake mediated by hSVCT1 was difficult to detect with this assay system (**Fig. L6**).

Fig. L6 | Scintillation count of transported C-14 radioisotope. (a) hSVCT1-overexpressing or **(b)** non-transfected HEK293T cells were exposed to buffer containing ^{14}C -radiolabeled ascorbate. Cells were collected immediately or after 30 minutes, and the lysates were subjected to scintillation counting. This experimental procedure is based on the successfully conducted colorimetric cellular assay.

This may be due to the endogenous expression of other transporters, such as GLUTs, which are known to transport the oxidized form of ascorbate. As ascorbate is prone to oxidation, the oxidized radiolabeled ascorbate may be imported into cells via these transporters. The linearity of the plots also supports this hypothesis. The oxidized ascorbate is detected together with the reduced form in RI-based assays, making it difficult to isolate and specifically detect the reduced form of ascorbate transported by hSVCT1 using this system.

Therefore, we concluded that liposome- or RI-based systems cannot be applied to the hSVCT1 substrate transport assay probably due to the intrinsic nature of this transporter, and decided to keep using the FRASC assay kit, which assesses the reduction activity of ascorbic acid. Several reports, including the mouse SVCT1 paper, have demonstrated that this assay kit and its chemical principles ensure the specific and quantitative measurements of only the reduced form of ascorbic acid. Using this strategy, we successfully obtained a Michaelis-Menten plot for each hSVCT1 mutant, including the additional ones in total of 20, and estimated their complete transport kinetics (**Fig. L7** and Supplementary Fig. 9).

Fig. L7 | Transport kinetics analysis of hSVCT1 mutants. Values are mean of $n = 3$ biological replicates \pm s.e.m. Asterisk indicates that the estimated K_m value exceeds the upper limit of the measurement range. All kinetics values and K_m curves were calculated using GraphPad Prism 9.5.1.

We apologize that we could not perform some of the proposed experiments. We spent much time struggling with the liposome and radioisotope trials, while the ITC and scintillation proximity assays were difficult due to the low yield of this protein. However, we added the

rigorous kinetics analysis of 20 hSVCT1 mutants in the revised manuscript, which we believe produced more reliable mechanistic insights into the ascorbate transport.

The assignment of the Na2 site is very interesting because it is different from previous reports of Na ion binding sites. Because it has not been so common to see a Na ion stabilized by an aromatic ring, it would be more rigorous if the authors can sustain their claims with a Na ion binding assay and mutations.

Thank you for the insightful comment. The residues mainly involved in the coordination of Na2 in our structure are S110 and F112 (**Fig. L1a**). Our structure suggests that Na2 is recognized by Phe112 via cation- π interaction. In addition to the mutational analysis of F112 in the original manuscript, we examined the transport activity of the S110A mutant. We measured the transport kinetics of both S110A and F112A mutants and compared them to that of the wild-type. The result shown in **Fig. L7** demonstrates that the transport activity is significantly reduced in both mutants, and the K_m value is largely increased, especially for the S110A mutant. The sidechain of S110 is distant from the ascorbate molecule and unable to interact directly with the ascorbate, but this result of the mutational analysis suggested the presence of a Na⁺ ion between S110 and the ascorbate, namely Na2, which contributes to the substrate recognition. Moreover, in the cryo-EM map of the occluded state, we observed density of a small molecule interacting with the amide of A113, where Na2 was assigned in the mouse SVCT1 structure. Considering that the occluded structure was obtained under Na⁺-free conditions, we assigned a water molecule at this site instead of Na⁺ (**Fig. L1c**), further supporting our assignment of Na2 in a different position.

Fig. L1 | Comparison of human and mouse SVCT1 density maps around the bound substrate. (a) Human SVCT1 density map (EMD-36201) and substrate-bound inward-open model (PDB 8JEW). **(b)** Mouse SVCT1 density map (EMD-34094) and substrate-bound inward-open model (PDB 7YTW). Both F_o maps are shown at the same contour level. **(c)** Water molecules in the substrate-free occluded state obtained under the Na^+ -free condition. Green mesh represents F_o-F_c omit map.

The observed structural change at the dimer interface is novel, and this is an aspect that has largely been ignored in the commonly invoked alternating access model for dimeric transporters. This is very exciting discovery, and it would catch more attention of the field if the authors could apply crosslinking to pinpoint interactions whose disruption or formation are required for substrate transport or binding.

Thank you for the positive comment. We strongly agree that this novel conformational state and transport mechanism are main points of this manuscript. Ideally, it would have been better to measure the transport activity after fixing the dimer interface by introducing cysteine mutants to cross-link each state. However, cross-linking is impossible because the substrate transported by this transporter is ascorbate, a reducing agent. Instead, we designed additional interface-disrupting mutants for each dimeric state and examined the relationship between dimer formation and transport activity (**Fig. L4**). These additional results also support our hypothesis of the new transport mechanism.

Fig. L4 | Dimer formation and transport activity of hSVCT1. (a–d) Close-up views of the residues responsible for dimer formation in the (a, b) inward-open state and (c, d) substrate-free occluded state. (e) Evaluation of oligomeric populations of dimer interface mutants by FSEC, detected by GFP fluorescence. (f) Substrate uptake assay for dimer interface mutants. Values are mean of $n = 3$ biological replicates \pm s.e.m. compared relative to the wild-type uptake rate.

Reviewer #3:

I have read the manuscript “Novel dimeric transport mechanism of human vitamin C transporter SVCT1” by Nureki and coworkers with great interest. It reports several cryo-electron microscopy (cryoEM) structures of human vitamin C transporter SVCT1 in different conformational states. The structures provide insights into conformational changes during substrate transport and identified critical residues participating in the substrate binding and transport. The work provides valuable insights into the conformational changes and critical residues involved in substrate binding and transport of SVCT1. However, I have a few major and minor comments that I believe will enhance the clarity and impact of your findings.

We thank the reviewer for the very positive comments. We have addressed the comments raised by the reviewer as follows.

MAJOR COMMENTS

1. My main concern is regarding the categorial processing using C2 symmetry. It would be crucial to attempt a reconstruction in C1 symmetry, and ideally perform symmetry expansion procedures, to investigate potential asymmetric states of the transporter. If there are conformational differences specific to each protomer, please speculate on the implications of these differences for vitamin transport.

Thank you for the insightful comments. For each of the substrate-bound inward-open and substrate-free occluded states, we applied symmetry expansion to the particle set of the stage before final classification (about 400,000 particles and contains some degree of non-homogeneity) and performed the subsequent analysis and reconstruction focusing on one protomer with C1 symmetry (**Fig. L13**).

The symmetry-expanded particles for the inward-open state were classified into two classes, and 3D reconstruction for each class resulted in 2.74 Å and 2.57 Å-resolution density maps. We built and refined atomic models for both maps, and compared the protomeric structures. The resulting RMSD value was 0.40 Å with good alignment, indicating that there was no protomeric asymmetry of the overall conformation in this state. We also compared the substrate binding sites, and although some of the residues were flipped, both structures bound a vitamin C molecule and two sodium ions, suggesting no asymmetry in the substrate binding. The particles for the substrate-free occluded state were analyzed in the same procedure, which yielded density maps of three classes with 2.87 Å, 3.21 Å, and 3.36 Å resolutions. The RMSD values between these classes are also small (0.21 Å between class 1 and class 2, 0.25 Å between class 1 and

class 3, and 0.23 Å between class 2 and class 3). Thus, we assume that the conformational change through the transport cycle of hSVCT1 follows a symmetric transition between two protomers.

Fig. L13 | Single-particle analysis of the symmetry-expanded datasets. (a) The C1-density maps for the substrate-bound inward-open particles. Single particle analysis was performed, focusing on the blue colored protomer for each class. (b) Density maps at the substrate binding pocket. (c) The C1-density maps for the substrate-free occluded particles. Single particle analysis was performed, focusing on the orange colored protomer for each class.

2. In the FSEC experiments, the presence of CHS with LMNG resulted in a shift towards dimer formation compared to the absence of CHS. In the cryoEM structures, endogenous cholesterol molecules are modeled at the interface between the gate and core domains, while CHS molecules are modeled at the dimer interface. On what basis were densities assigned to either cholesterol or CHS, and how confident are the authors in these assignments. Additionally, please speculate on the potential roles of cholesterol in dimer formation and the conformational dynamics between the core and gate domains?

Thank you for the comments. We assigned either cholesterol or a CHS molecule, based on the shape and size of the EM density maps and the surrounding chemical environment. The EM densities at the dimer interface clearly show both cholesteryl and hemisuccinate moieties, enabling us to confidently assign CHS molecules that are probably mimicking endogenous cholesterol molecules (**Fig. L14a–b**). For densities at the interface between the gate and core domains, we could observe the cholesteryl moiety, but we could not identify the hemisuccinate moiety from the densities themselves (**Fig. L14c**). Considering that the hydroxy or succinate moiety is likely oriented toward the protein core, rather than the lipid membrane, the hydroxy group of cholesterol fit well in the internal cavity, whereas the succinate moiety of CHS clashed with the surrounding residues. Thus, we assigned cholesterol molecules to these densities at the interface between the gate and core domains.

Fig L14 | CHS and cholesterol densities in the occluded structure. (a–b) The CHS models and $F_o - F_c$ omit map densities at the dimer interface. The hemisuccinate moieties are indicated by dotted circles. **(c)** The cholesterol model and $F_o - F_c$ omit map density at the interface between core and gate domains.

Regarding the potential roles of cholesterol molecules, we speculate that those at the dimer interface (where we assigned CHS molecules in this structure) in the substrate-free occluded state may stabilize the dimer formation, thereby facilitating the conformational change between the inward-open and substrate-free occluded states. Moreover, the movement of the gate domain is likely linked to the movement of the core domain. Therefore, cholesterol can indirectly contribute to the elevator mechanism and possibly to the subsequent transition to the outward-open state.

3. The substrate-binding pocket of SVCT1 is located at the end of a cleft between the gate and core domains, supported by an antiparallel β -strand sandwich. Please elaborate on the role of the β -strand sandwich in substrate binding and subsequent transport?

The β -strand region is sandwiched on both the extra- and intracellular sides by two short and oppositely directed α -helices, TM3 and TM10. This formation of the substrate binding pocket is typical for UraA fold transporters. The pocket region is surrounded by the opposite dipole moments formed by these short helices, and its electrostatic environment would be favorable for attracting negatively charged substrates, such as ascorbic acid. The β -strands themselves appear to be structurally required to form the substrate accommodation space inside the α helix-rich fold of the membrane transporter. We apologize for the misleading use of the word “sandwich” when describing the substrate binding pocket. We corrected the description in the revised manuscript.

4. The authors describe that the sodium 1 (Na1) binding site in human SVCT1 is similar to the mouse SVCT1 whereas Na2 site differs. It would be helpful to include an extended data figure comparing the substrate-binding sites of mouse and human SVCT1. Does the electrostatic potential surface of the binding sites offer any insights into differences in Na2 binding site? Also, does the mouse SVCT1 also display non-selectivity towards vitamin C and its isomers?

We thank the reviewer for the helpful suggestion. We have included supplementary figures comparing the substrate recognitions between human and mouse SVCT1 in Supplementary Fig. 7. The electrostatic potential surfaces of the substrate binding pockets of human and mouse SVCT1 are shown in **Fig. L15**, and no significant difference was observed between them. As compared to the mouse SVCT1 structure, we shifted the Na2 position based on our mutational results of Ser110 and Phe112, which recognize the sodium ion by cation- π and hydrogen-bonding interactions. Instead, we put a water molecule at the Na2 position in the mouse SVCT1. Although we could not find a previous study focusing on the substrate specificity between vitamin C and its isomers in mouse SVCT1, all studied mammalian SVCT1s show similar incomplete substrate specificity among vitamin C isomers, and the human and mouse SVCT1s were recently shown to non-specifically transport urate as well as ascorbate (Pflügers Arch. 475, 489–504, 2023). Considering that the residues involved in substrate recognition are similar among the human and mouse SVCT1s, mouse SVCT1 is likely to display the same non-selectivity toward vitamin C and its isomers.

Fig L15 | Comparison of the electrostatic surfaces of the human and mouse SVCT1 substrate binding pockets. The bound vitamin C molecule is shown as yellow sticks and Na² is shown as a violet sphere.

5. Through structural and functional studies, the authors showed that the dimer interface is important for substrate transport. Is there any cooperativity or influence of one protomer's substrate binding status on the second protomer's function?

As shown in **Fig. L13**, there is no asymmetry in the substrate's binding manner between protomers, and judging from this, the situation of substrate binding to only one protomer is unlikely to occur. Thus, the substrate transport by hSVCT1 is suggested to be based on a symmetric conformational change. This symmetric conformational change may have the advantage that the movement at the dimer interface can be more effectively amplified than when the conformational change occurs as a monomer, and thus dimer formation is important for the large-scale structural change we propose in the manuscript.

6. Based on the cryoEM structures of SVCT1 in different states, the authors propose a transport mechanism combining elevator and rotational motions-driven conformational transitions for substrate binding, transport, and release. It is an exciting claim for the transport of vitamin C through SVCT1. Do the authors have any additional theoretical or experimental evidence to support their transport mechanism?

Thank you for the positive comment. In the revised manuscript, we have designed several additional mutants of the residues at the dimer interface of each state. We have assessed their

dimer formation and transport activities (**Fig. L4**), and added the results in the revised manuscript. These additional results further support our hypothesis of the new transport mechanism.

Fig. L4 | Dimer formation and transport activity of hSVCT1. **a** Representations of dimer interfaces in the **(a)** inward-open state and **(b)** substrate-free occluded state, viewed from the intracellular side. **c–f** Close-up views of the residues responsible for dimer formation in **(c–d)** inward-open state and **(e–f)** substrate-free occluded state. **g** Evaluation of oligomeric population of dimer interface mutants by FSEC, detected by GFP fluorescence. **e** Substrate uptake assay for dimer interface mutants. Values are mean of $n = 3$ biological replicates \pm s.e.m. compared relative to the wild-type uptake rate.

7. Judging from the 2D classes it appears that the datasets show strong preferred orientation (mostly top views). Orientational distribution is not reported anywhere. Please update the Extended Data Figures with a representation and include a plot for 3DFSC figure of merit.

Thank you for the helpful comment. We have additionally calculated the orientational distribution of the particles in our structures using the 3DFSC processing server, and updated

the cryo-EM analysis workflow (**Fig. L16** and Supplementary Figs. 2–4). The 3DFSC sphericity value, which indicates the degree of anisotropy present in the structure, is approximately 0.9 or better for the three high-resolution structures mainly discussed in the manuscript, and hence there was no severe orientation bias.

Fig. L16 | 3DFSC analysis. (a) Substrate-bound inward-open state. (b) Substrate-free inward-open state. (c) Intermediate state. (d) Substrate-free occluded state. Plots were calculated by the Remote 3DFSC Processing Server.

8. Figures 1 and 3 should be merged into one figure to have a more direct comparison between the structures. The same is true for Figures 5 and 6, which should be merged.

Thank you for the helpful comment. We have rearranged the main figures accordingly.

MINOR COMMENTS

1. Line 181: The line reads, "substrate-binding pocket may be be too small to accommodate...". The word "be" is repeated twice.

We apologize for such a careless mistake. We fixed it in the revised manuscript.

2. Figure 1 legend, lines 415-416: "The core and gate domains of a protomer are colored light blue and deep blue, respectively." I suggest referring to the light blue color as "cyan" for clarity.

Thank you for the helpful comment. We fixed it in the revised manuscript.

3. In relation to minor comment 2, the Figure 5 legend mentions that the inward-open hSVCT1 structure is colored blue. I suggest referring to the blue color as "cyan" for clarity.

Reviewer #1 (Remarks to the Author):

The present manuscript is a revised version of the original manuscript entitled "Novel dimeric transport 1 mechanism of human vitamin C transporter SVCT1", by Kobayashi et al., which I have reviewed some time ago and asked for major experimental revision before being reconsidered for publication in Nat Comm. In particular, I asked authors to address specifically the functional and physiological relevance of some of the structural findings reported. The revised manuscript includes a significant number of the experiments asked, albeit not all, which definitely improve its rigorosity and better support some of the conclusions drawn.

The revised manuscript includes:

1. A more extensive mutational analysis of residues that seem to be involved in the two distinct dimeric states which support the importance of interactions proposed in the novel occluded conformation of hSVCT1. However, this is still a rather indirect proof of the novel occluded hSVCT1 structure proposed.
2. Cellular studies using GFP-tagged versions of the transporter mutated versions (all images of excellent quality), which show that in all cases hSVCT1 is properly translocated to the plasma membrane (PM) of HEK293T cells. This is an important addition as, together with experiments addressing the existence of intact monomers versus dimers, excludes the possibility that the mutations studied affect the gross folding of the transporter, which would have led to retention into the ER. Thus, these mutations might indeed have a negative specific effect the formation of the occluded conformation of hSVCT1 needed for transport catalysis. It is however interesting to note, and the authors should refer to this point, that the relative mutations do not affect translocation to the PM. In other words, monomers or aberrant dimers due to these mutations still find their way to the PM. A similar case has been reported in UapA, where mutations abolishing dimerization do not affect trafficking to the PM (Kourkoulou et al, 2019).
3. Uptake assays measuring transport rates in HEK293T cells expressing the wt or mutant version of hSVCT1. These have been performed using an enzymatic assay kit which measures accumulation of L-ascorbic acid in the cell indirectly. Although this kit is quite useful for measuring ascorbic acid transport in cells, it is not as rigorous as directly measuring the uptake and accumulation of radiolabeled L-ascorbic acid, which I have suggested. To their merit, the authors tried employing direct measurements of radiolabeled ascorbic acid accumulation. However, the low apparent radiolabeled substrate transport activity recorded in cells expressing hSVCT1 versions was undistinguishable from that detected in a negative control of non-transfected HEK293T cells. The authors speculate that this might be due to a background transport activity of oxidized L-ascorbic acid (i.e. dehydro-ascorbic acid) by GLUT-type transporters. I am afraid this is not a reasonable explanation. Why should their stock of L-ascorbic acid be oxidized after all? Even if a fraction of L-ascorbic is oxidized to dehydro-ascorbic acid they still should have seen a difference from the negative control. My guess is that in their radiolabeled substrate uptake measurements used a far too low concentration of 'hot' substrate (at the lowest μM range), which cannot be 'recognized' efficiently by hSVCT1, which is rather low/medium affinity L-ascorbic acid transporter. The authors do not describe in detail how they performed these radiolabeled substrate uptake assays.
4. Going back to the measurements the authors performed using the enzymatic kit, and in particular in the detailed transport kinetic assays presented in Figure L7 and Supplementary Fig. 9, I detect a very serious problem. To estimate K_m and V_m values, the concentrations of substrate used should cover a range below and above the presumed K_m of wt hSVCT1, which is close to 250 μM . The authors show only a single point (500 μM) exceeding the K_m of hSVCT1. A single deviation of measurements at this point would give a completely different estimation of K_m and V_m values. Furthermore, they report K_m values in mutants (e.g. F466A) where the transport activity is extremely low (e.g. 5%), which seems practically unfeasible. Furthermore, to my understanding, they also use uptake assays of 30 min, but report their results as transport rates per 1 min, considering that transport is linear for 1-30 min. This might be true for the wt, but in the mutants is not known. Consequently, most of the mutations analyzed appear to have a moderate negative effect of 3-4-fold, at max, in respect to K_m values. In conclusion, the transport kinetics analysis of hSVCT1 mutants shown in Fig. L7 and Supplementary Fig. 9 are not valid as they are, and should thus be repeated, or removed from any manuscript considered for publication.

5. The authors also tried to perform a liposome assay using radiolabeled ascorbate to measure the transport kinetics of the mutants, but this approach failed.
6. The authors did not try to perform fluorescence assays, as I suggested, to show directly whether the mutations studied indeed affect dimer formation in the occluded structure.
7. The authors investigated the relationship between solubilization conditions and monomer-dimer formation of hSVCT1 and found that a simple alteration of detergents (LMNG to DDM) hinders dimer formation and results in larger monomer populations in micelles (Fig. L2). This is an interesting and instructive finding supporting the flexibility of dimer formation in different hydrophobic environments, which further highlights the structural and functional significance of transporter-lipid interactions in vivo.
8. In Fig. L8 they re-examine the oligomeric populations for N230A-related mutants and find a very different result, better fit to their speculations, from that reported in the original manuscript. How can this be possible?
9. Based on my comments on their structural analysis, they also recalculate segments of their structures and improve their resolution.

Reviewer #2 (Remarks to the Author):

The authors made substantial effort to address my comments on the flexible dimerization interface and substrate/Na binding sites. Although the experiments that I wished to see did not happen, which leaves room for alternative interpretations to the structural observations, I remain overall positive to the study.

Reviewer #3 (Remarks to the Author):

I congratulate the authors to this exciting manuscript. I have no further comments.

Reviewer #1:

The present manuscript is a revised version of the original manuscript entitled “Novel dimeric transport mechanism of human vitamin C transporter SVCT1”, by Kobayashi et al., which I have reviewed some time ago and asked for major experimental revision before being reconsidered for publication in Nat Comm. In particular, I asked authors to address specifically the functional and physiological relevance of some of the structural findings reported. The revised manuscript includes a significant number of the experiments asked, albeit not all, which definitely improve its rigorousness and better support some of the conclusions drawn.

We appreciate the reviewer for positively considering the revision of the manuscript. We have addressed the additional comments raised by the reviewer as follows:

The revised manuscript includes:

- 1. A more extensive mutational analysis of residues that seem to be involved in the two distinct dimeric states which support the importance of interactions proposed in the novel occluded conformation of hSVCT1. However, this is still a rather indirect proof of the novel occluded hSVCT1 structure proposed.*
- 2. Cellular studies using GFP-tagged versions of the transporter mutated versions (all images of excellent quality), which show that in all cases hSVCT1 is properly translocated to the plasma membrane (PM) of HEK293T cells. This is an important addition as, together with experiments addressing the existence of intact monomers versus dimers, excludes the possibility that the mutations studied affect the gross folding of the transporter, which would have led to retention into the ER. Thus, these mutations might indeed have a negative specific effect the formation of the occluded conformation of hSVCT1 needed for transport catalysis. It is however interesting to note, and the authors should refer to this point, that the relative mutations do not affect translocation to the PM. In other words, monomers or aberrant dimers due to these mutations still find their way to the PM. A similar case has been reported in UapA, where mutations abolishing dimerization do not affect trafficking to the PM (Kourkoulou et al, 2019).*

We thank the reviewer for providing additional helpful comments. Following the suggestion, we have added the text mentioning to this point.

- 3. Uptake assays measuring transport rates in HEK293T cells expressing the wt or mutant version of hSVCT1. These have been performed using an enzymatic assay kit which measures*

accumulation of L-ascorbic acid in the cell indirectly. Although this kit is quite useful for measuring ascorbic acid transport in cells, it is not as rigorous as directly measuring the uptake and accumulation of radiolabeled L-ascorbic acid, which I have suggested. To their merit, the authors tried employing direct measurements of radiolabeled ascorbic acid accumulation. However, the low apparent radiolabeled substrate transport activity recorded in cells expressing hSVCT1 versions was undistinguishable from that detected in a negative control of non-transfected HEK293T cells. The authors speculate that this might be due to a background transport activity of oxidized L-ascorbic acid (i.e. dehydro-ascorbic acid) by GLUT-type transporters. I am afraid this is not a reasonable explanation. Why should their stock of L-ascorbic acid be oxidized after all? Even if a fraction of L-ascorbic is oxidized to dehydro-ascorbic acid they still should have seen a difference from the negative control. My guess is that in their radiolabeled substrate uptake measurements used a far too low concentration of 'hot' substrate (at the lowest μM range), which cannot be 'recognized' efficiently by hSVCT1, which is rather low/medium affinity L-ascorbic acid transporter. The authors do not describe in detail how they performed these radiolabeled substrate uptake assays.

Thank you for the comments. We apologize for not describing the RI assay method in detail. The process was as follows: HEK293T cells were cultured in DMEM medium supplemented with 10% (v/v) FBS, at 37°C with 5% CO₂. Cells were seeded in 6-well culture plates at a density of 4×10^5 cells/mL and transiently transfected with 1 μg of pEG BacMam vectors containing the hSVCT1 wild-type coding region using the Lipofectamine 3000 transfection reagent, then cultured for 48 h at 37°C with 5% CO₂. ¹⁴C-labeled ascorbic acid (American Radiolabeled Chemicals) was mixed with non-radiolabeled sodium ascorbate at the ratio of hot : cold = 1 : 49, and this mixture was used for the preparation of radiolabeled substrate-containing buffer with different concentrations (0, 10, 20, 50, 100, 200, and 500 μM). Culture medium was replaced with 1 mL of radiolabeled ascorbate-containing buffer. Cells were harvested immediately (as the sample of 0 min) or after incubating for 30 min at 37°C (as the sample of 30 min), washed twice, and solubilized for 30 min with 500 μL of 10% Triton X-100. Insoluble materials in the cell lysates were removed by centrifugation at 12,000 rpm for 20 min, then 100 μL aliquots of supernatant were applied to filter paper and dried. The dried filter paper was soaked into 5 mL of Ultima Gold liquid scintillation cocktail (Perkin Elmer) in a glass scintillation vial (Fischer Scientific), and scintillation count was conducted for 1 min using AccuFLEX LSC-8000 scintillation counter (Hitachi). However, the detected count was quite low (**Fig. L1a**), so in the next trial, we increased the hot ratio by 4-fold (i.e. the ratio of hot : cold = 4 : 46) and decreased the solubilization scale to 100 μL in order to raise the concentration of hot substrate in the cell lysate. Nevertheless, the counts were still small and difficult to distinguish

from those of mock cells, thus we could not detect the hSVCT1-related substrate uptake effectively (Fig. L1b–c).

Fig. L1 | Scintillation count trials of ¹⁴C-radiolabeled ascorbate. (a) Initial trial with the hot ratio of 1/50. **(b–c)** Second trial with the increased hot ratio of 4/50 for hSVCT1 transfected of mock cells.

Our speculation that L-ascorbic acid was oxidized is based on the relatively unstable nature of L-ascorbic acid, which can be readily oxidized by high temperature or light (Yin *et al.*, 2022), and the abundant expression of GLUTs, which mediate nonspecific transport of dehydroascorbic acid (Medina *et al.*, 2002). However, as you pointed out, it may be unlikely that L-ascorbic acid was oxidized to an undetectable level in such a short period of time, and the major reason for the difficulty in scintillation count might be the low concentration of radiolabeled substrate. Exactly as you suggested, the ratio of hot substrate was very low due to the scarce amount of available radiolabeled compound. Therefore, the inefficient recognition of radiolabeled ascorbic acid by hSVCT1 due to the low concentration may be the reason why the RI assay was not successful.

4. Going back to the measurements the authors performed using the enzymatic kit, and in particular in the detailed transport kinetic assays presented in Figure L7 and Supplementary Fig. 9, I detect a very serious problem. To estimate K_m and V_m values, the concentrations of substrate used should cover a range below and above the presumed K_m of wt hSVCT1, which is close to 250 µM. The authors show only a single point (500 µM) exceeding the K_m of hSVCT1. A single deviation of measurements at this point would give a completely different estimation of K_m and V_m values. Furthermore, they report K_m values in mutants (e.g. F466A) where the transport activity is extremely low (e.g. 5%), which seems practically unfeasible. Furthermore, to my understanding, they also use uptake assays of 30 min, but report their results as transport

rates per 1 min, considering that transport is linear for 1-30 min. This might be true for the wt, but in the mutants is not known. Consequently, most of the mutations analyzed appear to have a moderate negative effect of 3-4-fold, at max, in respect to Km values. In conclusion, the transport kinetics analysis of hSVCT1 mutants shown in Fig. L7 and Supplementary Fig. 9 are not valid as they are, and should thus be repeated, or removed from any manuscript considered for publication.

We thank the reviewer for the comments. As you pointed out, there were indeed some inadequacies in the kinetics analysis of the mutants. Following the reviewer's suggestion, we decided to remove Supplementary Fig. 9 from the manuscript.

- 5. The authors also tried to perform a liposome assay using radiolabeled ascorbate to measure the transport kinetics of the mutants, but this approach failed.*
- 6. The authors did not try to perform fluorescence assays, as I suggested, to show directly whether the mutations studied indeed affect dimer formation in the occluded structure.*
- 7. The authors investigated the relationship between solubilization conditions and monomer-dimer formation of hSVCT1 and found that a simple alteration of detergents (LMNG to DDM) hinders dimer formation and results in larger monomer populations in micelles (Fig. L2). This is an interesting and instructive finding supporting the flexibility of dimer formation in different hydrophobic environments, which further highlights the structural and functional significance of transporter-lipid interactions in vivo.*
- 8. In Fig. L8 they re-examine the oligomeric populations for N230A-related mutants and find a very different result, better fit to their speculations, from that reported in the original manuscript. How can this be possible?*

We thank the reviewers for the comments. The FSEC evaluation was performed for each trial of ascorbate uptake assay, and although the tendency of dimer formation is generally the same, there was some variation in the oligomer ratio and the shape of the peaks in each trial. Comparing the N230A-related curves used in the original manuscript with the curves from other trials, the dimer ratio of the N230A single mutant in this trial was found to be higher than those commonly observed in other trials (**Fig. L2**). Thus, after considering the results from multiple trials, the most typical and representative result was instead used in the revised manuscript.

Fig. L2 | All FSEC evaluations of dimer formation of N230A-related mutants. Results of trials 2 was used in the original manuscript, and that of trial 5 is used in the revised manuscript.

9. Based on my comments on their structural analysis, they also recalculate segments of their structures and improve their resolution.

Thank you again for all the constructive and insightful comments. We have addressed the comments raised by the reviewer, and hope that it is now acceptable for publication.

Reviewer #2:

The authors made substantial effort to address my comments on the flexible dimerization interface and substrate/Na binding sites. Although the experiments that I wished to see did not happen, which leaves room for alternative interpretations to the structural observations, I remain overall positive to the study.

Thank you for the positive comments on our revised manuscript, and we apologize that we could not perform all the suggested experiments in the revision.

Reviewer #3:

I congratulate the authors to this exciting manuscript. I have no further comments.

Thank you for the kind review on our revised manuscript.

Reviewer #1 (Remarks to the Author):

My concerns on the manuscript entitled "Novel dimeric transport mechanism of human vitamin C transporter SVCT1" have been addressed in the revisions at a significant degree.